# Geometric Decoupling: Diagnosing the Structural Instability of Latent

Yuanbang Liang [* 1]   Zhengwen Chen [* 1]   Yu-Kun Lai [1]

## Abstract

Latent Diffusion Models (LDMs) achieve high-fidelity synthesis but suffer from latent space brittleness, causing discontinuous semantic jumps during editing. We introduce a Riemannian framework to diagnose this instability by analyzing the generative Jacobian, decomposing geometry into *Local Scaling* (capacity) and *Local Complexity* (curvature). Our study uncovers a **"Geometric Decoupling"**: while curvature in normal generation functionally encodes image detail, Out-of-Distribution (OOD) generation exhibits a functional decoupling where extreme curvature is wasted on unstable semantic boundaries rather than perceptible details. This geometric misallocation identifies "Geometric Hotspots" as the structural root of instability, providing a robust intrinsic metric for diagnosing generative reliability. Our code is at https://github.com/Byronliang8/Diffusion-Geometry.

## 1. Introduction

Latent Diffusion Models (LDMs) (Song et al., 2020a; Rombach et al., 2021; Podell et al., 2023; Esser et al., 2024; Labs et al., 2025) have fundamentally reshaped the landscape of generative AI, achieving unprecedented fidelity and diversity by decoupling perceptual compression from semantic generation. However, this generative prowess conceals a critical structural flaw: the **instability of the latent space**. While LDMs excel at sampling from a distribution, they are notoriously fragile when tasked with traversing it (Kwon et al., 2022; Guo et al., 2024). Minor perturbations to the latent code often result in discontinuous semantic jumps, shattering the smoothness required for controlled editing (Kwon et al., 2022; Tumanyan et al., 2023), interpolation (Song et al., 2020b; Guo et al., 2024), and inversion (Mokady et al., 2023; Wallace et al., 2023).

---
*Equal contribution [1]School of Computer Science and Informatics, Cardiff University, Cardiff, United Kingdom. Correspondence to: Yu-Kun Lai <LaiY4@cardiff.ac.uk>.

*Proceedings of the 43rd International Conference on Machine Learning*, Seoul, South Korea. PMLR 306, 2026. Copyright 2026 by the author(s).

Why does a model capable of generating photorealistic intricacies fail to maintain semantic continuity over microscopic distances? Existing literature typically attributes this to the general non-linearity of deep networks. However, these explanations remain qualitative. They fail to quantify where the manifold breaks, why the editing directions diverge, and what geometric cost the model pays to achieve its high-fidelity output.

In this work, we employ a Riemannian geometric lens (Sakai, 1996; Lebanon, 2002; Lee, 2018) to investigate these questions. Specifically, we utilize the metrics of Local Scaling (LS), which measures information capacity via volume expansion, and Local Complexity (LC), which measures geometric curvature and directional stability. While these metrics have been established in broader manifold learning contexts, their behavior within the specialized latent spaces of LDMs, particularly under semantic stress, remains unexplored.

Our research uncovers a profound phenomenon within LDMs, which we term **Geometric Resource Misallocation**. We define this as the process where the model's geometric budget, specifically its structural curvature (LC) and volume expansion (LS), is forcibly redirected away from encoding perceptually meaningful details and toward resolving irreconcilable semantic constraints. Through a rigorous comparative analysis, we distinguish between In-Distribution (normal) samples and confirmed Out-of-Distribution (OOD) generations. Crucially, we observe that OOD prompts do not always trigger a departure from the natural image manifold; however, in instances where the generative process does yield an OOD image (*e.g.* structural hallucinations), we identify a critical fracture in the model's geometric logic. Under normal generation, we observe a functional coupling where LC correlates with high-frequency detail of the local tangent vector (called Projected High-Frequency Energy, PHFE which will be defined later), implying that the manifold curves purposefully to encode complex features.

In stark contrast, under semantic stress (OOD), this relationship collapses. While LS remains a robust predictor of image detail, the correlation between LC and PHFE drops significantly. We term this phenomenon the **"Geometric Decoupling"** of LDMs: to reconcile conflicting semantic constraints, the model forces the principal editing direction

to rotate instantaneously, incurring extreme curvature costs that are functionally decoupled from actual detail generation. In these OOD regions, high curvature is no longer an efficient encoding mechanism but a pathological byproduct of semantic conflict.

This paper provides the first quantitative diagnosis of this trade-off, shifting the paradigm from viewing LDM instability as a random artifact to understanding it as a structural cost of semantic generalization. Our specific contributions are as follows:

- Identification of Geometric Functional Decoupling: We provide the first empirical evidence that the functional role of geometric curvature (LC) in LDMs is context-dependent. We demonstrate a "correlation gap" where LC's coupling with image detail (PHFE) collapses under OOD conditions, revealing that OOD instability is driven by non-functional manifold twisting.

- Quantitative Characterization of "Geometric Decoupling": We characterize the quantitative manifestation of Geometric Decoupling by demonstrating that OOD generations trigger an abnormal surge in both LC and LS magnitudes. Furthermore, we show that while the model drastically increases its geometric effort to handle semantic stress, the generative efficiency of this effort, operationally defined by the capacity of LC to encode perceptible detail (PHFE), diminishes severely.

- Interpolation Trajectory Analysis on Manifold Instability: We conduct an Interpolation Trajectory Analysis to provide a dynamic perspective on manifold instability. We demonstrate that Out-of-Distribution (OOD) conditions induce pathological tortuosity and heavy-tailed discontinuities, validating that the geometric decoupling observed locally translates to severe structural brittleness during latent traversal.

## 2. Background and Related Work

**The Geometry of Deep Generative Models.** Deep generative models are fundamentally viewed as mappings that embed a low-dimensional latent manifold into a high-dimensional observation space. To understand the structural properties of this embedding, foundational works have utilized the Jacobian matrix $\mathbf{J} = \partial G / \partial \mathbf{z}$ as a linear approximation of the local geometry.

Seminal works by Shao et al. (2018) and Chen et al. (2019) introduced Riemannian metrics to measure geodesic distances within the latent space, arguing that standard linear interpolation in $\mathcal{Z}$ is suboptimal due to manifold curvature. This Jacobian-based framework has since been extensively applied across various architectures to quantify curvature,

capacity, and expressivity. Specifically, geometric analysis has been established for VAEs (Chadebec & Allassonniere, 2022; Galperin & Köthe, 2024; Lee & Park, 2023), Normalizing Flows (Caterini et al., 2021), and GANs (Dahal et al., 2022; Dai & Hang, 2021).

While recent studies have begun to explore the geometry of diffusion models (Park et al., 2023; Tang & Yang, 2024; Kamkari et al., 2024; Farghly et al., 2025) and Large Language Models (LLMs) (He et al., 2026; Wang et al., 2026), these works primarily focus on trajectory curvature or adaptation. Most relevant to our work, Humayun et al. (2024) and Humayun et al. (2025) formalized the notions of *Local Scaling* (LS) and *Local Complexity* (LC) to explicitly quantify the trade-off between generative capacity and manifold curvature. However, the behavior of these geometric descriptors within the iterative, multi-stage latent space of LDMs, especially under Out-of-Distribution (OOD) conditions, remains an open question.

**Out-of-Distribution & Hallucination in Generative AI.** Evaluating the reliability of generative models requires analyzing their behavior not just at the mode of the distribution (normal generation), but also in the low-density regions corresponding to Out-of-Distribution (OOD) samples. In the context of image synthesis, OOD samples, often manifested as hallucinations, structural anomalies, or artifacts, represent a departure from the learned data manifold.

Prior research has primarily approached OOD detection through statistical or perceptual lenses. Likelihood-based methods (Song et al., 2017; Nalisnick et al., 2018; Choi et al., 2018; Kirichenko et al., 2020) attempt to identify outliers via density estimation, while reconstruction-based approaches (Sakurada & Yairi, 2014; Zhou & Paffenroth, 2017; Zong et al., 2018; Zenati et al., 2018) measure the deviation of samples from a learned prior. In recent research within the diffusion and LLM/VLM domains, methods often rely on monitoring the internal attention maps (Prabhakaran et al., 2025; Binkowski et al., 2025; Oorloff et al., 2025) or understanding interpolation failure modes (Aithal et al., 2024).

However, these methods treat the OOD status as a binary or scalar property, overlooking the underlying geometric mechanism that produces these samples. Our work provides a structural definition of OOD generation. We posit that OOD images are not merely statistical outliers but are the product of specific **geometric pathologies** within the latent mapping. By characterizing OOD regions through the decoupling of Local Scaling and Local Complexity, we offer a metric that explains *how* the model's geometric logic fractures when generating content outside its training distribution.

**Latent Space Instability and Geometric Trade-offs.** The latent space of LDMs serves as the operational substrate

for controlled generation, enabling applications such as semantic editing (Hertz et al., 2022), style transfer (Zhang et al., 2023), and concept personalization (Ruiz et al., 2023).

However, empirical evidence contradicts the assumption of global smoothness necessary for these tasks. Mokady et al. (2023) observed that standard inversion often fails to preserve fidelity, necessitating trajectory rectification, while Kwon et al. (2022) identified inherent irregularities in the noisy latent space ($z$-space) compared to feature spaces. While these works propose heuristic solutions to mitigate instability, they treat the manifold's roughness as a black-box phenomenon.

From a theoretical perspective, this instability reflects a fundamental tension between model expressivity and geometric stability. Classical deep learning theory suggests that high local curvature is a necessary cost for modeling high-capacity distributions (Bartlett et al., 2017; Miyato et al., 2018; Jordan & Dimakis, 2021). Ideally, generative manifolds obey a "Law of Parsimony," allocating geometric distortion strictly to regions of high semantic density, as observed in generative models (Humayun et al., 2024). Our work bridges these empirical observations and theoretical frameworks by providing a **diagnostic quantification**. Unlike heuristic fixes, we identify a "Geometric Decoupling" in diffusion models: the manifold exhibits *pathological curvature*, incurring high geometric costs with low informational gain, thereby challenging the assumption that neural networks inherently learn the most efficient representation of the data manifold.

## 3. Riemannian Diagnosis of Latent Manifolds

To quantitatively diagnose the structural instability of Latent Diffusion Models, we propose a geometric framework that decomposes the generative mapping into distinct properties of capacity, curvature, and functional content. Let $G : \mathcal{Z} \to \mathcal{X}$ denote the generator, mapping a latent code $\mathbf{z} \in \mathbb{R}^E$ to the data space $\mathbf{x} \in \mathbb{R}^{D_{\text{output}}}$.

### 3.1. Subspace Jacobian Approximation

Direct computation of the full Jacobian $\mathbf{J} \in \mathbb{R}^{D_{\text{output}} \times E}$ is computationally intractable due to the high dimensionality of the image space. We employ a matrix-free finite difference approach projected onto a low-dimensional subspace. Let $\mathbf{W} \in \mathbb{R}^{E \times P}$ be an orthonormal projection matrix defining a random subspace of dimension $P \ll E$.

We approximate the subspace Jacobian $\mathbf{J}_{\text{sub}} \in \mathbb{R}^{D_{\text{output}} \times P}$ column-wise. The $i$-th column, corresponding to the perturbation direction $\mathbf{w}_i$, is computed as:

$$\mathbf{J}_{\text{sub}} \cdot [\mathbf{w}_i] \approx \frac{G(\mathbf{z} + \epsilon \mathbf{w}_i) - G(\mathbf{z})}{\epsilon} \tag{1}$$

where $\epsilon$ is a sufficiently small perturbation radius.

From this, we construct the local metric tensor $\mathbf{A} \in \mathbb{R}^{P \times P}$, which encapsulates the local geometry of the manifold within the subspace:

$$\mathbf{A} = \mathbf{J}_{\text{sub}}^{\text{T}} \mathbf{J}_{\text{sub}} \tag{2}$$

By performing eigendecomposition $\mathbf{A} = \mathbf{V} \boldsymbol{\Lambda} \mathbf{V}^{\text{T}}$, we obtain the eigenvalues $\boldsymbol{\Lambda} = \text{diag}(\lambda_1, \ldots, \lambda_P)$ and eigenvectors $\mathbf{V} = [\mathbf{v}_1, \ldots, \mathbf{v}_P]$, which serve as the basis for our geometric descriptors.

### 3.2. Geometric Descriptors of the Manifold

Following the previous research about local complexity and local scaling (Hanin & Rolnick, 2019; Wang et al., 2020; Patel & Montufar, 2025; Humayun et al., 2024; 2025), we isolate two distinct geometric properties from the spectral components of the metric tensor. For more discussion details about Geometric Descriptors, please check Appendix B.2.

**Local Scaling (LS).** LS measures the **local information capacity** via the volume expansion rate of the manifold. It is defined as the log-sum of singular values $\sigma_i = \sqrt{\lambda_i}$:

$$\psi_{\boldsymbol{\omega}}(\mathbf{z}) = \frac{1}{2} \sum_{i=1}^{P} \log(\lambda_i) \cdot \mathbf{1}_{\{\lambda_i > 0\}} \tag{3}$$

High $\psi_{\boldsymbol{\omega}}$ indicates a region where the latent volume is significantly expanded, theoretically allowing for the encoding of dense information.

**Local Complexity (LC).** LC measures the **geometric instability** or curvature of the manifold. It focuses on the principal direction of change, $\mathbf{V}_1(\mathbf{z})$ (the eigenvector corresponding to the largest eigenvalue $\lambda_{\max}$). LC is defined as the rate of rotation of $\mathbf{V}_1$ within a local neighborhood $\mathcal{N}_{\epsilon}(\mathbf{z})$:

$$\delta(\mathbf{z}) = \mathbb{E}_{\mathbf{z}' \sim \mathcal{N}_{\epsilon}(\mathbf{z})} \left[ \frac{\|\mathbf{V}_1(\mathbf{z}) - \mathbf{V}_1(\mathbf{z}')\|_2}{\|\mathbf{z} - \mathbf{z}'\|_2} \right] \tag{4}$$

High $\delta$ implies that the direction of maximum semantic change rotates rapidly, leading to unstable traversal trajectories.

### 3.3. Diagnosing Geometric Functionality: $\mathbf{P}_1$ and PHFE

To determine whether the curvature ($\delta$) is functionally useful or redundant, we analyze the content encoded along the principal axis.

**Principal Direction Projection ($\mathbf{P}_1$).** We define $\mathbf{P}_1$ as the projection of the latent principal axis $\mathbf{V}_1$ onto the data space via the Jacobian-Vector Product (JVP):

$$\mathbf{P}_1 = \mathbf{J}_{\text{sub}} \mathbf{V}_1 \tag{5}$$

Mathematically, $\mathbf{P}_1$ represents the instantaneous rate of change of the image $\mathbf{x}$ along $\mathbf{V}_1$. This can be derived from the total differential $d\mathbf{x} = \mathbf{J}d\mathbf{z}$. For a perturbation $\Delta\mathbf{z} = \epsilon\mathbf{V}_1$, the output change approaches the directional derivative:

$$\lim_{\epsilon \to 0} \frac{G(\mathbf{z} + \epsilon\mathbf{V}_1) - G(\mathbf{z})}{\epsilon} \approx \mathbf{J}_{\text{sub}}\mathbf{V}_1 = \mathbf{P}_1 \qquad (6)$$

Thus, $\mathbf{P}_1$ visualizes the exact visual content being altered by the model's most unstable geometric direction. Please check appendix. B.3 for more computation details.

**Projected High-Frequency Energy (PHFE)** To diagnose the functional utility of the manifold's curvature, we analyze the content encoded along the principal axis using PHFE.

It is critical to distinguish our proposed PHFE from the standard High-Frequency Energy (HFE) of the generated image.

- **HFE** measures the static detail of the image $\mathbf{x}$: $\text{HFE}(\mathbf{z}) = \text{Var}(\nabla^2\mathbf{x})$. It indicates whether the generated image contains high-frequency features.

- **PHFE** measures the dynamic detail of the tangent vector $\mathbf{x}_{\text{proj}}$. It indicates whether the *change* in the image (driven by manifold curvature) is high-frequency or low-frequency.

We apply the Laplacian operator ($\nabla^2$) to $\mathbf{x}_{\text{proj}}$ to extract the high-frequency components of the instantaneous change:

$$\mathbf{x}_{\text{proj}}^{\text{laplace}} = \nabla^2\mathbf{x}_{\text{proj}} \qquad (7)$$

where $\mathbf{x}_{\text{proj}}^{\text{laplace}}$ is the image representing the second-order pixel variations (edges, textures) driven by the $\mathbf{V}_1$ direction.

We define the Projected High-Frequency Energy (PHFE) as the variance of the Laplacian response applied to the change vector $\mathbf{x}_{\text{proj}}$:

$$\text{PHFE}(\mathbf{z}) = \text{Var}\left(\nabla^2\mathbf{x}_{\text{proj}}\right) = \text{Var}\left(\nabla^2(\mathbf{J}_{\text{sub}}\mathbf{V}_1)\right) \qquad (8)$$

This metric serves as a crucial diagnostic probe:

- **Functional Coupling:** If $\delta$ correlates with PHFE, the high curvature is functionally necessary to encode complex high-frequency details.

- **Geometric Decoupling:** If $\delta$ is high but PHFE is low, the curvature is wasted on non-semantic distortions or low-frequency global shifts, indicating a geometric pathology.

## 3.4. Spectral Structure and Dimensionality

To investigate the structural dimensionality of the latent manifold under semantic stress, we perform spectral analysis on the local metric tensor $\mathbf{A}$ and introduce two specific metrics.

**Spectral Isolation Score (SIS).** Grounded in matrix perturbation theory (Davis & Kahan, 1970), which posits that the stability of an eigenvector is determined by its separation from the rest of the spectrum, we introduce SIS to quantify the isolation of the principal direction from the secondary semantic subspace:

$$\text{SIS} = \frac{\text{CosSim}(\mathbf{V}_1, \mathbf{v}_1')}{\sum_{k=2}^{P} \text{CosSim}(\mathbf{V}_1, \mathbf{v}_k')} \qquad (9)$$

where $\mathbf{V}_1$ is the principal eigenvector at $\mathbf{z}$ and $\{\mathbf{v}_k'\}$ is the eigenbasis at a perturbed neighbor $\mathbf{z}'$.

A higher SIS indicates a **"Tunnel Vision"** geometry, where the manifold locks rigidly onto a single dominant axis while severing connections to secondary semantic dimensions. This mathematically formalizes the pathology of *Dimensionality Collapse* (Jing et al., 2021; Gao et al., 2019), where high-dimensional spaces degenerate into rigid, narrow cones. By exhibiting extreme variation along only one direction while discarding others, the model loses the structural degrees of freedom necessary to synthesize coherent and diverse image details. For more information about matrix perturbation theory and Spectral Isolation Score (SIS), please check Appendix. B.5.

**Dimensional Coupling Ratio ($\phi_{\text{dim}}$).** To compare the structural degradation between Out-of-Distribution (OOD) and Normal conditions, we define the Dimensional Coupling Ratio for the $k$-th axis as the relative loss of coupling strength:

$$\rho_{\text{dim}}^{(k)} = 1 - \frac{\text{CosSim}(\mathbf{V}_1, \mathbf{v}_k')_{\text{OOD}}}{\text{CosSim}(\mathbf{V}_1, \mathbf{v}_k')_{\text{Normal}}} \qquad (10)$$

This metric quantifies the fraction of secondary coupling lost due to semantic stress.

## 3.5. Interpolation Trajectory Analysis

To quantify geometric stability beyond local neighborhoods, we analyze how a diffusion sampler maps a smooth interpolation in the *noise* space to an *induced* trajectory in the final $\mathbf{x_0}$ latent space under different prompt conditions.

**Noise-space interpolation.** For each trial, we sample two i.i.d. noise latents $\mathbf{z}_A, \mathbf{z}_B \sim \mathcal{N}(\mathbf{0}, \mathbf{I})$ and construct a spherical linear interpolation (slerp), $\mathbf{z}(\alpha) = \text{slerp}(\mathbf{z}_A, \mathbf{z}_B; \alpha), \alpha \in [0, 1]$. We discretize $\alpha$ into $K$ uniform steps $\{\alpha_k\}_{k=0}^{K}$ and obtain $\{\mathbf{z}(\alpha_k)\}_{k=0}^{K}$.

**Model-induced $\mathbf{x_0}$ latent trajectory.** Let $\Phi_c(\cdot)$ denote the diffusion sampling map under prompt condition $c$ (including

guidance and all sampling hyperparameters), which takes an initial noise latent to the final $\mathbf{x_0}$ latent. For each $\alpha_k$, we run the sampler initialized at $\mathbf{z}(\alpha_k)$ and record the resulting final latent: $\mathbf{h}_c(\alpha_k) = \Phi_c\big(\mathbf{z}(\alpha_k)\big)$, where $\mathbf{h}_c(\alpha_k)$ is the latent at the last denoising step (*i.e.*, the predicted $\mathbf{x_0}$ latent). Crucially, the same noise endpoints $(\mathbf{z}_A, \mathbf{z}_B)$ are shared across conditions to enable paired Normal vs. OOD comparisons.

**Trajectory discretization in $\mathbf{x_0}$ latent space.** We define the stepwise jump magnitude on the induced trajectory: $\Delta_k^{(c)} = \|\mathbf{h}_c(\alpha_{k+1}) - \mathbf{h}_c(\alpha_k)\|_2$, and $k = 0, \ldots, K-1$.

**Geometric path metrics.** Using $\{\Delta_k^{(c)}\}$, we quantify trajectory roughness and efficiency in the final latent space:

Cumulative Path Length ($L$): $L^{(c)} = \sum_{k=0}^{K-1} \Delta_k^{(c)}$.

Endpoint Distance ($D$): $D^{(c)} = \|\mathbf{h}_c(\alpha_K) - \mathbf{h}_c(\alpha_0)\|_2$.

Tortuosity ($\tau$): $\tau^{(c)} = \frac{L^{(c)}}{D^{(c)} + \varepsilon}$.

Excess Length ($E$): $E^{(c)} = L^{(c)} - D^{(c)}$.

A larger $\tau^{(c)}$ or $E^{(c)}$ indicates that a smooth noise-space interpolation is mapped to a more irregular and inefficient traversal in the final $\mathbf{x_0}$ latent space under condition $c$.

**Extremal Trajectory Increments.** While $L^{(c)}$, $\tau^{(c)}$ and $E^{(c)}$ summarize the *overall* inefficiency of the induced trajectory $\{\mathbf{h}_c(\alpha_k)\}_{k=0}^{K}$, we further detect *localized discontinuities* by analyzing the **extremal statistics** of the stepwise increments $\{\Delta_k^{(c)}\}_{k=0}^{K-1}$, where $\Delta_k^{(c)} = \|\mathbf{h}_c(\alpha_{k+1}) - \mathbf{h}_c(\alpha_k)\|_2$.

Let $Q_q(\cdot)$ denote the $q$-quantile operator. We define the upper-tail quantile increment as

$$\Delta^{q,(c)} := Q_q\Big(\{\Delta_k^{(c)}\}_{k=0}^{K-1}\Big),$$

where $q \in (0, 1)$, so that, *e.g.* $\Delta^{0.95,(c)}$ is the 95th percentile (only the largest 5% of increments exceed it). We report $\Delta^{0.90,(c)}$ and $\Delta^{0.95,(c)}$, together with the maximum increment, $\Delta^{\max,(c)} := \max_{0 \le k \le K-1} \Delta_k^{(c)}$.

Across paired Normal/OOD evaluations using the *same noise endpoint pair* $(\mathbf{z}_A, \mathbf{z}_B)$ (and the same discretization and sampler configuration), we summarize the consistency of an OOD increase for any metric $m$ by the *probability-of-increase*,

$$\mathrm{frac}(m) := \Pr\big(m_{\mathrm{OOD}} > m_{\mathrm{Normal}}\big),$$

where trials are indexed by noise endpoint pairs and $N$ is the number of paired samples. Intuitively, large $\Delta^{0.95,(c)}$ and $\Delta^{\max,(c)}$ indicate rare but severe local "shocks" along the induced $\mathbf{x_0}$-latent trajectory.

## 4. Experiments

**Experiment Setup.** We empirically investigate the functional relationship between manifold geometry and generated image content. We hypothesize that in a stable generative process, local curvature (LC) should be functionally coupled with the encoding of high-frequency detail. To test this, we constructed a paired dataset of $N = 500$ generated samples using fixed random seeds across two conditions: (1) **Normal**, using standard semantic prompts; and (2) **OOD**, using structurally anomalous prompts. For each sample, we computed Local Complexity (LC), Local Scaling (LS), and Projected High-Frequency Energy (PHFE).

**Generative Models.** We test our Riemannian Diagnosis with Stable Diffusion 3.5 Medium (SD3.5) (Esser et al., 2024) and FLUX.1 (Labs et al., 2025).

**Hyper-parameters.** Unless otherwise specified, we adhere to the standard inference configurations prescribed for each pre-trained model. Specifically, we utilize the default number of denoising steps (*e.g.*, 50 steps for DDIM) to ensure that our geometric analysis reflects the model's behavior under typical operating conditions. For the comparative analysis between OOD and normal prompts, we generate and evaluate a sample set of 100 images for each category.

**OOD Setup.** We define an OOD prompt as a photorealistic description that combines a subject drawn from the COCO object set (Lin et al., 2014). Our OOD focuses on realized OOD generations where the anomalous constraint is visibly manifested, for an OOD prompt does not always produce an OOD image. Paired Normal/OOD prompts share the same subject, composition, and lighting, and differ only in the contradiction clause. For more information please check Appendix E and the example of prompts in the Table 15.

### 4.1. The Correlation Gap

We investigate the functional relationship between manifold geometry and generated image content. To ensure robustness across diverse semantic contexts, we aggregated a total pool of 900 samples for each condition (Normal and OOD) using varied prompts. From this pool, we performed 10 independent subsampling runs, randomly selecting $N = 500$ samples per condition for each iteration to calculate the Spearman Rank Correlation ($\rho$) between latent geometry (LC, LS) and visual content (PHFE).

Table 1 summarizes the correlation analysis averaged over 10 subsampling runs. The data reveals a stark dichotomy in geometric functionality. Take SD3.5 as an example, and similar observations also apply to Flux.1. For Normal prompts, both LS and LC exhibit positive correlations with image detail ($\rho(\mathrm{LS}, \mathrm{PHFE}) \approx 0.84$; $\rho(\mathrm{LC}, \mathrm{PHFE}) \approx 0.41$). However, under OOD conditions, while LS remains a robust predictor of detail ($\rho \approx 0.82$), the correlation between LC

and PHFE collapses to negligible levels ($\rho \approx 0.08$).

*Table 1.* Spearman correlations between geometric measures and encoded detail for Stable Diffusion 3.5 and Flux.1. The significant drop in $\rho(\mathrm{LC}, \mathrm{PHFE})$ for OOD samples quantifies the geometric decoupling. The correlations for different prompts please check Appendix C. For more extended statistical and geometric validations, please check Appendix D. Normal: Normal prompts, OOD: out-of-distribution prompts

*(a)* Stable Diffusion 3.5

|  | $\rho(\mathrm{LC}, \mathrm{PHFE})$ | $\rho(\mathrm{LS}, \mathrm{PHFE})$ | $\rho(\mathrm{LS}, \mathrm{LC})$ |
|---|---|---|---|
| Normal | $0.413 \pm 0.036$ | $0.836 \pm 0.016$ | $0.606 \pm 0.023$ |
| OOD | $0.083 \pm 0.027$ | $0.824 \pm 0.011$ | $0.296 \pm 0.018$ |

*(b)* Flux.1

|  | $\rho(\mathrm{LC}, \mathrm{PHFE})$ | $\rho(\mathrm{LS}, \mathrm{PHFE})$ | $\rho(\mathrm{LS}, \mathrm{LC})$ |
|---|---|---|---|
| Normal | $0.448 \pm 0.014$ | $0.827 \pm 0.024$ | $0.612 \pm 0.031$ |
| OOD | $0.269 \pm 0.021$ | $0.780 \pm 0.009$ | $0.395 \pm 0.078$ |

The persistence of the LS-PHFE correlation across both conditions confirms that Local Scaling serves as a consistent proxy for information capacity; volume expansion reliably translates to visual detail regardless of semantic validity. Crucially, the collapse of the LC-PHFE correlation in OOD samples ($\Delta\rho \approx -0.33$) provides quantitative evidence of **Geometric Decoupling**. In OOD regions, the model expends extreme geometric curvature that is no longer functionally utilized to encode perceptible details.

## 4.2. High-Frequency Transfer Collapse

To test whether latent geometric high-frequency potential translates into perceptible image detail under semantic stress, we quantify the **Transfer Efficiency** as

$$\eta = \frac{\mathrm{HFE}_{\mathrm{image}}}{\mathrm{PHFE}_{\mathrm{latent}}}, \qquad \mathrm{HFE}_{\mathrm{image}} = \mathrm{Var}\left(\nabla^2 \mathbf{x}\right), \quad (11)$$

where $\mathrm{Var}(\nabla^2 \mathbf{x})$ measures image-space high-frequency energy and $\mathrm{PHFE}_{\mathrm{latent}}$ measures the projected high-frequency energy in the latent manifold. Using paired evaluation with identical random seeds (same prompt-pair, same seed), we compare Normal vs. OOD and compute the differential shift $\Delta\eta = \eta_{\mathrm{OOD}} - \eta_{\mathrm{Normal}}$ via paired bootstrap ($N = 500$).

**Top$k$-HF (HF concentration).** Let $\mathbf{x} \in \mathbb{R}^{H \times W \times 3}$ be the generated RGB image and define the Laplacian magnitude map $m_{ij} = \frac{1}{3} \sum_{c=1}^{3} \left|(\nabla^2 \mathbf{x}_c)_{ij}\right|$. Let $\Omega$ be the set of all pixel indices and $\Omega_k$ the subset containing the top $k\%$ indices with largest $m_{ij}$. We define the high-frequency concentration

$$\mathrm{Top}k\text{-HF} = \frac{\sum_{(i,j) \in \Omega_k} m_{ij}}{\sum_{(i,j) \in \Omega} m_{ij} + \varepsilon}, \quad (12)$$

where a lower value indicates more spatially diffuse (noise-like) high-frequency patterns.

As shown in Table 2, OOD prompts trigger a substantial increase in latent geometric energy ($\mathrm{PHFE}_{\mathrm{latent}}$ rises from $158.506$ to $252.799$, $\sim 59.5\%$), while the image-space high-frequency energy remains nearly unchanged ($\mathrm{Var}(\nabla^2 \mathbf{x})$: $0.0120 \rightarrow 0.0131$). Crucially, under the same stochastic conditions (same seed), switching from Normal to OOD therefore increases the latent "cost" without a commensurate gain in perceptible high-frequency output, implying a **drop in transfer efficiency** $\eta$. Consistent with a noise-like manifestation, the high-frequency concentration decreases under OOD (Top5-HF: $0.530 \rightarrow 0.493$; Top10-HF: $0.691 \rightarrow 0.663$), indicating that the produced high-frequency patterns become more spatially diffuse. Higher-$k$ variants (Top15/Top20) exhibit the same trend but with smaller magnitude, suggesting a robust shift in high-frequency morphology rather than an isolated threshold effect; for more related Top-$k$ results, please check Table 17 in appendix H.

*Table 2.* Median statistics for high-frequency transfer. OOD samples exhibit a latent energy surge ($\mathrm{PHFE}_{\mathrm{latent}}$) that does not translate into image-space high-frequency energy ($\mathrm{Var}(\nabla^2 \mathbf{x})$). Under paired seeds, this implies reduced transfer efficiency $\eta = \mathrm{HFE}_{\mathrm{image}}/\mathrm{PHFE}_{\mathrm{latent}}$ and more spatially diffuse high-frequency patterns (lower Top$k$-HF).

|  | $\mathrm{PHFE}_{\mathrm{latent}}$ | $\mathrm{Var}(\nabla^2 \mathbf{x})$ | Top5-HF | Top10-HF |
|---|---|---|---|---|
| Normal | 158.506 | 0.0120 | 0.530 | 0.691 |
| OOD | 252.799 | 0.0131 | 0.493 | 0.663 |

## 4.3. Manifold Rigidity and Dimensional Collapse

Table 3 presents the spectral coupling profile across three semantic categories. The data reveals a consistent structural degradation in OOD manifolds, characterized by the decoupling of secondary axes. For the *Chair* prompt (structural violation), we observe a drastic collapse: $\mathrm{sim}(\mathbf{V}_1, \mathbf{v}_2')$ drops from $0.140$ to $0.084$, corresponding to $\rho_{\mathrm{dim}} \approx 0.40$ (a $40\%$ loss in coupling). For *Fisher* and *Penguin* prompts, the drop is consistent around $17\%$. This indicates that under OOD stress, the manifold loses its "width," severing the orthogonal connections required for flexible semantic editing.

Correspondingly, the SIS increases across all OOD conditions, peaking at $6.45$ for the Chair prompt ($+31\% \uparrow$). This quantitative shift confirms the transition to a **Hyper-Rigid** state: the generative trajectory becomes locked into a single, isolated principal direction, explaining the "stiffness" and lack of correctability observed in hallucinatory generations.

## 4.4. Qualitative Representative Selection

We provide visual evidence of "Pathological Energy Distributions" by spatially mapping the geometric instability onto the generated image domain. This heatmap visualizes the spatial distribution of the manifold's local sensitivity.

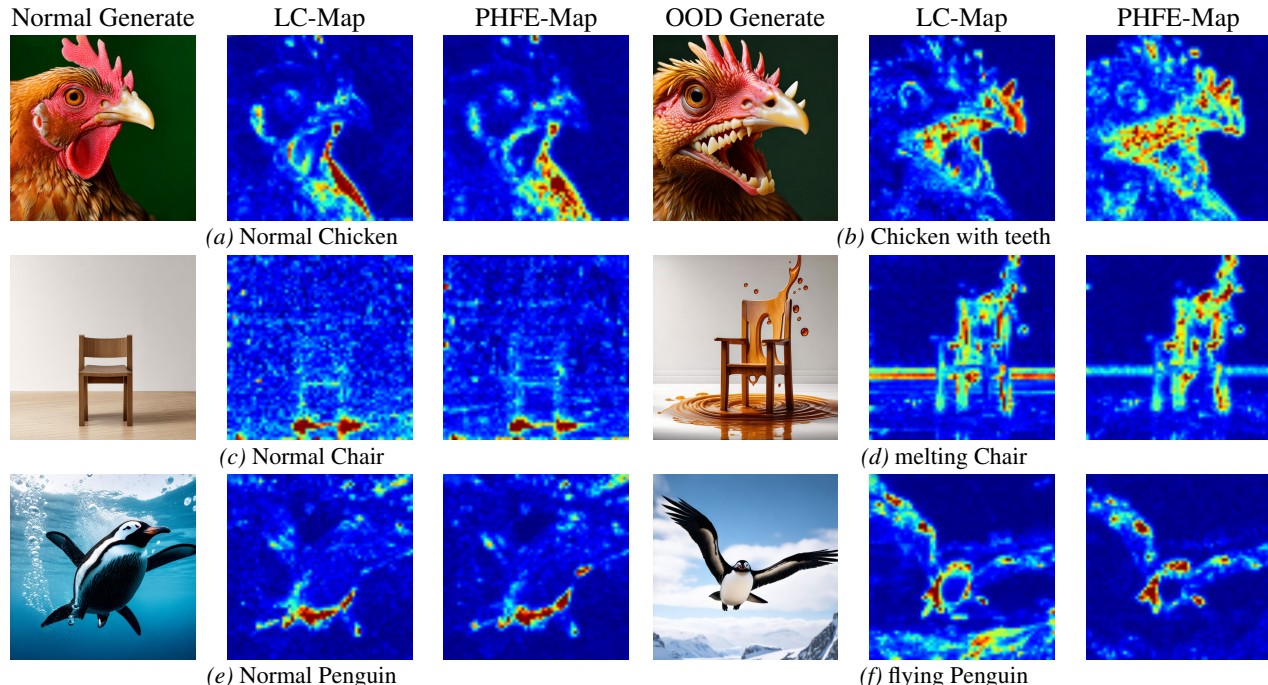

| Normal Generate | LC-Map | PHFE-Map | OOD Generate | LC-Map | PHFE-Map |

*(a)* Normal Chicken  *(b)* Chicken with teeth

*(c)* Normal Chair  *(d)* melting Chair

*(e)* Normal Penguin  *(f)* flying Penguin

*Figure 1.* **Qualitative Visualization of Geometric Decoupling.** We display Normal (a,c,e) and OOD (b,d,f) samples alongside their Local Complexity Maps (LC-Map) and Projected High-Frequency Energy Maps (PHFE-Map). In each subfigure, from left to right: the Generated Image, the LC-Map, and the PHFE-Map. Red regions in the LC-Map denote "Geometric Hotspots" of extreme curvature. This spatial correspondence confirms that the model allocates its maximum geometric complexity to resolving semantic conflicts, often decoupling from the actual high-frequency detail (PHFE), thereby illustrating the misallocation of geometric resources. The more examples please check Appendix. F.1.

*Table 3.* **Spectral Coupling Profile ($\mathcal{S}_k$) and Dimensional Collapse.** We report the cosine similarity between the perturbed principal vector $\mathbf{V}_1$ and the original eigenbasis $[\mathbf{v}'_2, \ldots, \mathbf{v}'_{min}]$ for Normal vs. OOD conditions. Across all prompts, OOD samples exhibit a systematic collapse in secondary coupling (positive $\phi_{dim}$) and a corresponding surge in Spectral Isolation Score (SIS), confirming that hallucinations induce a rigid, lower-dimensional manifold geometry. $\phi_{dim}$ represents the Dimensional Coupling Loss, where higher values indicate severe decoupling. SIS represents Spectral Isolation Score, and higher SIS indicates a "Tunnel Vision" geometry, where the manifold locks rigidly onto a single axis while severing connections to secondary semantic dimensions.

*(a)* Normal Fish vs Fish with legs.

| | $(\mathbf{V}_1, \mathbf{v}'_2)$ | $(\mathbf{V}_1, \mathbf{v}'_3)$ | $(\mathbf{V}_1, \mathbf{v}'_4)$ | $(\mathbf{V}_1, \mathbf{v}'_5)$ | $(\mathbf{V}_1, \mathbf{v}'_{min})$ | SIS |
|---|---|---|---|---|---|---|
| Normal | 0.150±0.100 | 0.066±0.047 | 0.045±0.035 | 0.034±0.029 | 0.013±0.014 | 4.365±1.976 |
| OOD | 0.125±0.066 | 0.055±0.033 | 0.038±0.025 | 0.029±0.021 | 0.011±0.010 | 5.044±2.119 |
| $\phi_{dim}$ | 0.172 | 0.162 | 0.154 | 0.151 | 0.172 | |

*(b)* Normal Chair vs Melting Chair.

| | $(\mathbf{V}_1, \mathbf{v}'_2)$ | $(\mathbf{V}_1, \mathbf{v}'_3)$ | $(\mathbf{V}_1, \mathbf{v}'_4)$ | $(\mathbf{V}_1, \mathbf{v}'_5)$ | $(\mathbf{V}_1, \mathbf{v}'_{min})$ | SIS |
|---|---|---|---|---|---|---|
| Normal | 0.140±0.097 | 0.062±0.046 | 0.042±0.035 | 0.032±0.030 | 0.012±0.014 | 4.909±2.338 |
| OOD | 0.084±0.056 | 0.040±0.027 | 0.027±0.019 | 0.020±0.014 | 0.007±0.005 | 6.453±1.828 |
| $\phi_{dim}$ | 0.401 | 0.355 | 0.357 | 0.151 | 0.417 | |

*(c)* Normal Penguin vs Flying Penguin.

| | $(\mathbf{V}_1, \mathbf{v}'_2)$ | $(\mathbf{V}_1, \mathbf{v}'_3)$ | $(\mathbf{V}_1, \mathbf{v}'_4)$ | $(\mathbf{V}_1, \mathbf{v}'_5)$ | $(\mathbf{V}_1, \mathbf{v}'_{min})$ | SIS |
|---|---|---|---|---|---|---|
| Normal | 0.103±0.075 | 0.046±0.041 | 0.031±0.031 | 0.024±0.026 | 0.008±0.010 | 6.393±2.206 |
| OOD | 0.085±0.031 | 0.037±0.015 | 0.024±0.010 | 0.019±0.008 | 0.006±0.003 | 6.614±1.768 |
| $\phi_{dim}$ | 0.175 | 0.196 | 0.226 | 0.208 | 0.250 | |

As illustrated in Figure 1, the LC-Map exhibits intense activation "hotspots" (highlighted in red) precisely where the generative prior conflicts with the prompt constraints:

**Biological Anomaly:** In the case of a "chicken with teeth," the LC-Map shows peak curvature concentrated exclusively on the beak region, indicating extreme manifold distortion required to synthesize the unnatural dental features.

**Material Conflict:** For the "melting chair," geometric stress is localized to the liquifying structural components, where the rigid prior of the object clashes with the fluid dynamics of the prompt.

**Functional Hybridization:** In the "flying penguin" sample, the manifold instability is mapped directly to the wings, reflecting the model's struggle to reconcile the anatomical constraints of a flightless bird with the semantic requirement of flight.

These visualizations confirm that "Geometric Decoupling" is a localized phenomenon: the model expends its maximum geometric budget (highest curvature) specifically to enforce the most counter-factual semantic attributes. The more examples, generated by Flux.1 and stable diffusion 3.5, please check Appendix. F.1.

## 4.5. Interpolation Trajectory Analysis

To evaluate geometric stability during latent traversal, we first construct spherical linear interpolation in the *initial noise space* between randomly sampled endpoints $(\mathbf{z}_A, \mathbf{z}_B)$, yielding $\mathbf{z}(t_k)$ on a fixed grid. For each $\mathbf{z}(t_k)$, we run the sampler under condition $c \in \{\text{Normal}, \text{OOD}\}$ to obtain the corresponding terminal $\mathbf{x_0}$-latent, denoted by $\mathbf{h}_c(t_k)$. All trajectory metrics below are computed on the induced $\mathbf{x_0}$-latent path $\{\mathbf{h}_c(t_k)\}_{k=0}^K$.

*Table 4.* Interpolation trajectory metrics (Monte Carlo mean $\pm$ std across trials). We report absolute values under Normal/OOD, followed by the ratio $R_m = \frac{m_{\text{OOD}}}{m_{\text{Normal}}}$. Each trial samples $n=100$ seed pairs with stratification, over 50 trials.

|        | $L$             | $\tau$          | $E$             |
| ------ | --------------- | --------------- | --------------- |
| Normal | $5.374 \pm 0.044$ | $5.695 \pm 0.040$ | $4.428 \pm 0.039$ |
| OOD    | $6.243 \pm 0.033$ | $6.446 \pm 0.031$ | $5.295 \pm 0.038$ |
| $R$    | $1.183 \pm 0.009$ | $1.150 \pm 0.008$ | $1.219 \pm 0.010$ |

*Table 5.* Monte-Carlo estimates of the trajectory increments at each interpolation step $k$. We report $\Delta$ (OOD-Normal) and the ratio (OOD/Normal). OOD trajectories show a systematic increase in step size across the entire path. Please check Appedix. G for all steps. The Mean in the table is averaged over all steps.

| $k$  | Normal         | OOD            | OOD-Normal      | OOD/Normal      |
| ---- | -------------- | -------------- | --------------- | --------------- |
| 0    | $0.252\pm0.011$ | $0.301\pm0.012$ | $0.032\pm0.018$ | $1.100\pm0.057$ |
| 5    | $0.261\pm0.011$ | $0.315\pm0.012$ | $0.049\pm0.021$ | $1.147\pm0.066$ |
| 10   | $0.255\pm0.010$ | $0.306\pm0.012$ | $0.057\pm0.019$ | $1.176\pm0.063$ |
| 15   | $0.249\pm0.011$ | $0.308\pm0.012$ | $0.069\pm0.018$ | $1.226\pm0.066$ |
| 19   | $0.258\pm0.012$ | $0.297\pm0.011$ | $0.038\pm0.020$ | $1.118\pm0.065$ |
| Mean | **0.259**      | **0.304**      | **0.0431**      | **1.133**       |

### 4.5.1. GLOBAL PATH ANALYSIS: INEFFICIENCY AND EXPANSION

We first evaluate the global geometric structure by analyzing both the aggregate path metrics (Table 4) and the step-wise increment consistency (Table 5).

**Pathological Tortuosity:** OOD trajectories exhibit significant geometric inefficiency. The Cumulative Path Length ($L$) increases by 18.3%, and Tortuosity ($\tau$) increases by 15.0% relative to Normal paths. This implies the induced $\mathbf{x_0}$-latent transport is more tortuous / less contractive.

**Systematic Manifold Expansion:** This inefficiency is not isolated to specific segments but is systemic. As shown in Table 5, the mean trajectory increment is elevated across all interpolation steps (ratio $> 1.13$). The global mean increment rises from 0.2585 (Normal) to 0.3036 (OOD), a relative expansion of $\sim 17.4\%$.

**Geometric Excess:** Critically, while the Euclidean Endpoint Distance ($D$) increases only mildly ($R_D \approx 1.038$), the Excess Length ($E = L - D$) surges by 21.9%. This decoupling suggests that OOD prompts do not merely push latent points further apart but actively **warp the space between them**, significantly increasing the geometric cost of traversal.

### 4.5.2. ANALYSIS OF LOCAL DISCONTINUITIES

Finally, we analyze the statistics of **Extremal Trajectory Increments** (Table 6) to detect abrupt semantic shifts.

*Table 6.* Metrics for **Extremal Trajectory Increments** quantifying local discontinuities. The increase in $\Delta^{0.95}$ and $\Delta^{\max}$ confirms that OOD paths exhibit abrupt transitions. frac $= \Pr(m_{\text{OOD}} > m_{\text{Normal}})$; $R_m = \frac{m_{\text{OOD}}}{m_{\text{Normal}}}$.

|        | $\Delta^{0.90}$   | $\Delta^{0.95}$   | $\Delta^{\max}$   |
| ------ | ----------------- | ----------------- | ----------------- |
| Normal | $0.406 \pm 0.003$ | $0.453 \pm 0.004$ | $0.530 \pm 0.005$ |
| OOD    | $0.450 \pm 0.003$ | $0.496 \pm 0.003$ | $0.568 \pm 0.004$ |
| $R$    | $1.133 \pm 0.010$ | $1.118 \pm 0.008$ | $1.110 \pm 0.010$ |
| frac   | $0.728$           | $0.680$           | $0.568$           |

**Extremal Increments.** Let $\Delta_k^{(c)} = \|\mathbf{h}_c(t_{k+1}) - \mathbf{h}_c(t_k)\|_2$ denote the per-segment jump along the induced $\mathbf{x_0}$-latent trajectory. Following Sec. 3.5, we summarize the upper tail of $\{\Delta_k^{(c)}\}$ by $\Delta^{0.90,(c)}$, $\Delta^{0.95,(c)}$, and $\Delta^{\max,(c)} = \max_k \Delta_k^{(c)}$.

**Sign consistency.** We additionally report the sign-consistency statistic frac $= \Pr(m_{\text{OOD}} > m_{\text{Normal}})$ (defined in Sec. 3.5), frac $= 0.680$ indicates a consistent tendency toward larger upper-tail jumps under OOD (above chance). Such extremal increments typically coincide with rapid, non-smooth semantic transitions in the generated sequence, consistent with a brittle OOD manifold.

*Table 7.* Impact of Training Intervention on Geometric Coupling.

| Model | $\rho(\text{LC}, \text{PHFE})_{\text{Normal}}$ | $\rho(\text{LC}, \text{PHFE})_{\text{OOD}}$ |
|---|---|---|
| SD3.5 Base | 0.413 | 0.083 |
| SD3.5 Turbo | 0.342 | 0.156 |

## 5. Investigating the Causal Nature of Geometric Decoupling

To address whether the observed geometric decoupling is a correlational artifact or a structural consequence of the generative process, we conducted a controlled, training-level intervention. We compared two models with identical architectures but distinct training regimes: Stable Diffusion 3.5 (SD3.5) Base and SD3.5 Turbo.

As shown in Table 7 and Table 8, this intervention yielded two key findings that provide meaningful causal evidence:

**Sensitivity to Training Regimes:** ADD distillation (Sauer et al., 2024) measurably reduces the decoupling gap, shrinking the correlation drop from $-80\%$ to $-54\%$. A purely correlational artifact would likely remain invariant across different training methodologies.

**Structural Asymmetry:** The intervention selectively affects the coupling between LC and PHFE, while leaving the relationship between LS and PHFE statistically stable.

Crucially, this asymmetric pattern replicates across distinctly different model architectures. As detailed in Table 8, the $\Delta$ LC-PHFE varies significantly depending on the architecture and training pipeline, whereas the $\Delta$ LS-PHFE remains consistently marginal.

While definitive causal proof necessitates full retraining with explicit curvature constraints, the metric's sensitivity to training interventions combined with cross-architecture replication provides strong evidence that geometric decoupling is a fundamental structural mechanism rather than a mere statistical coincidence.

## 6. Validated Practical Applications of the Geometric Metric

Beyond theoretical diagnostics, our geometric framework provides validated, zero-retraining utility across the full Latent Diffusion Model (LDM) deployment lifecycle. We outline two distinct practical applications below.

### 6.1. Annotation-Free OOD Detection

We evaluated whether geometric variables can autonomously detect Out-of-Distribution (OOD) generation failures without human annotation. Testing on 500 Normal and 500 OOD samples using SD3.5, we compared the raw

*Table 8.* Cross-Architecture Asymmetric Replication. The correlation drop ($D$) represents the relative percentage change in Spearman's $\rho$ when transitioning from Normal to OOD generation, calculated as $D = \frac{\rho_{\text{OOD}} - \rho_{\text{Normal}}}{\rho_{\text{Normal}}}$.

| Model | Architecture | $D$(LC-PHFE) | $D$(LS-PHFE) |
|---|---|---|---|
| FLUX.1 Base | Hybrid DiT | $-40\%$ | $-5.7\%$ |
| SD3.5 Turbo | MMDiT | $-54\%$ | $-0.6\%$ |
| SD3.5 Base | MMDiT | $-80\%$ | $-1.5\%$ |

LC metric, the raw LS metric, and our proposed geometric efficiency ratio (LC/PHFE).

As shown in Table 9, neither LC nor LS can reliably detect OOD states in isolation. However, the geometric efficiency ratio achieves an AUROC of 0.816 requiring zero ground-truth annotation. This validates the core decoupling hypothesis: OOD stress does not systematically shift absolute LC values, but it destroys the functional relationship between LC and PHFE.

*Table 9.* OOD Detection Performance ($N = 1000$, SD3.5). LC/PHFE directly operationalizes geometric decoupling as a per-image anomaly score; neither LC (Curvature) nor LS (Capacity) alone achieves reliable detection.

| Score | AUROC | Results |
|---|---|---|
| LS (raw) | 0.199 | fails |
| LC (raw) | 0.427 | fails |
| **LC/PHFE (ours)** | **0.816** | **works** |

### 6.2. Distillation and Training Monitoring

Because the correlation drop is highly sensitive to training-regime changes (improving from $-80\%$ to $-54\%$ under ADD distillation) while the baseline capacity metric remains invariant, it serves as a powerful monitoring signal. Developers can track geometric coupling during distillation or fine-tuning to measure structural integrity, providing an architecture-agnostic quality signal that traditional pixel-space metrics like FID or CLIP score simply cannot capture.

## 7. Conclusion

We introduce a Riemannian framework to diagnose LDM instability, revealing a fundamental **"Geometric Decoupling."** While Local Scaling (Capacity) efficiently drives fidelity, OOD stress triggers a pathological surge in Local Complexity (Curvature) that functionally decouples from image detail. This **Geometric Resource Misallocation**, where extreme curvature is wasted on unstable boundaries rather than perceptible features, identifies the structural root of interpolation failure and semantic discontinuities. Please check Appendix I for the limitations and future work.

## Acknowledgements

We appreciate the reviewers' constructive comments in improving the paper. This research was funded by the UK Engineering and Physical Sciences Research Council (No. EP/Y028805/1).

## Impact Statement

This paper presents work whose goal is to advance the field of Machine Learning. There are many potential societal consequences of our work, none which we feel must be specifically highlighted here.

However, broadly speaking, our work contributes to the trustworthiness and reliability of generative AI. By identifying the geometric signatures of hallucinations and Out-of-Distribution (OOD) generation, our proposed framework provides a rigorous, intrinsic method for detecting model failures before they manifest as visual errors. This is particularly critical for deploying diffusion models in safety-sensitive domains (e.g., medical imaging or autonomous simulation), where structural consistency is paramount. Furthermore, our diagnostic metrics (Local Complexity and PHFE) offer a pathway to "Geometric-Aware" auditing tools, enabling developers to quantify the stability of latent spaces and mitigate the risks of unpredictable semantic jumps in automated content generation pipelines.

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

## A. Theoretical Analysis of OOD Geometric Stress

A potential concern in evaluating Out-of-Distribution (OOD) performance is the finite nature of human-defined anomalous prompts. However, we argue that the observed *Geometric Decoupling* is not a function of prompt quantity, but a structural consequence of the **Manifold Mismatch Hypothesis**.

### A.1. The Semantic-Geometric Coupling Principle

Let $\mathcal{Z} \subset \mathbb{R}^d$ be the latent manifold and $G : \mathcal{Z} \to \mathcal{X}$ be the generative mapping. For a "Normal" prompt, the model's training objective enforces a Lipschitz-continuous mapping where local curvature $\delta$ is minimized to maintain semantic stability. Mathematically, the energy required to encode detail is bounded:

$$\|\nabla G(z)\| \propto \mathcal{S}(z, p) \tag{13}$$

where $\mathcal{S}$ is the semantic alignment between latent code $z$ and prompt $p$.

### A.2. Curvature as a Measure of Semantic Stress

Under OOD conditions, the prompt $p_{ood}$ exists in a region of the text-embedding space where the conditional distribution $P(x|p_{ood})$ is sparsely populated or non-existent in the training data. To satisfy the cross-attention mechanism, the model must "force" the latent code into a configuration that satisfies contradictory structural constraints.

From a Riemannian perspective, this "forcing" manifests as an **instantaneous rotation of the principal Jacobian direction**. If even a single OOD prompt induces a transition from a low-curvature regime to a high-curvature regime while simultaneously decoupling from the output energy (PHFE), it provides sufficient evidence of a *local structural breakdown*.

### A.3. Generalization via Local Manifold Properties

Because Local Complexity (LC) and Local Scaling (LS) are intrinsic properties of the Jacobian $J_z = \frac{\partial G}{\partial z}$, our findings suggest that OOD prompts act as "probes" that uncover pre-existing **Geometric Hotspots** in the LDM latent space.

- **Universality:** If the model is a smooth approximator, these hotspots are not isolated artifacts but are indicative of the "stiffness" of the manifold when departing from the training distribution.

- **Sufficiency of Representative Cases:** By analyzing "Confirmed Hallucinations" (via seed selection), we isolate the exact moments where the manifold's curvature becomes non-functional. This "case-study" approach is mathematically robust because a single counter-example of geometric efficiency is sufficient to disprove the global stability of the LDM manifold.

### A.4. Discussion on Experimental Scope

While the number of OOD prompts is finite, the diversity of seeds tested ($N = 500$) ensures that we are sampling a statistically significant volume of the local latent neighborhood. The consistency of the "Correlation Gap" across these samples suggests that the observed *Geometric Decoupling* is a fundamental byproduct of the LDM's architecture rather than a prompt-specific anomaly.

## B. Detailed Methodology and Mathematical Derivations

In this appendix, we provide rigorous mathematical formulations for the geometric descriptors used in our analysis. We detail the subspace approximation of the Jacobian, the spectral decomposition of the local metric tensor, and the derivation of the Principal Direction Projection ($\mathbf{P}_1$) as a differential rate of semantic change.

### B.1. Core Computational Flow

#### B.1.1. FD-Subspace Jacobian Matrix $\mathbf{J}_{\text{SUB}}$

Direct computation of the full Jacobian $\mathbf{J} \in \mathbb{R}^{D_{\text{output}} \times E}$ is computationally intractable for Latent Diffusion Models due to the high dimensionality of both the latent space $E$ and the image space $D_{\text{output}}$. To overcome this, we employ a **Finite Difference Subspace** approach.

*Table 10.* Spearman correlations between geometric measures and encoded detail with different prompts on Stable Diffusion 3.5. The significant drop in $\rho(\text{LC}, \text{PHFE})$, $\rho(\text{LS}, \text{PHFE})$ and $\rho(\text{LS}, \text{LC})$ for OOD samples quantifies the geometric decoupling. Normal: Normal prompts, OOD: out-of-distribution prompts

*(a)* chair vs melting chairs

|  | $\rho(\text{LC}, \text{PHFE})$ | $\rho(\text{LS}, \text{PHFE})$ | $\rho(\text{LS}, \text{LC})$ |
|---|---|---|---|
| Normal | $0.389 \pm 0.113$ | $0.823 \pm 0.0062$ | $0.616 \pm 0.087$ |
| OOD | $-0.069 \pm 0.142$ | $0.782 \pm 0.078$ | $0.043 \pm 0.119$ |

*(b)* Spoon vs jelly liked Spoon

|  | $\rho(\text{LC}, \text{PHFE})$ | $\rho(\text{LS}, \text{PHFE})$ | $\rho(\text{LS}, \text{LC})$ |
|---|---|---|---|
| Normal | $0.549 \pm 0.110$ | $0.747 \pm 0.080$ | $0.781 \pm 0.083$ |
| OOD | $0.140 \pm 0.155$ | $0.701 \pm 0.099$ | $0.510 \pm 0.120$ |

*(c)* Cat vs large human-like eye Cat

|  | $\rho(\text{LC}, \text{PHFE})$ | $\rho(\text{LS}, \text{PHFE})$ | $\rho(\text{LS}, \text{LC})$ |
|---|---|---|---|
| Normal | $0.546 \pm 0.075$ | $0.785 \pm 0.067$ | $0.755 \pm 0.052$ |
| OOD | $0.156 \pm 0.133$ | $0.567 \pm 0.118$ | $0.413 \pm 0.114$ |

Let $\mathbf{W} \in \mathbb{R}^{E \times P}$ be an orthonormal projection matrix defining a random lower-dimensional subspace of dimension $P \ll E$. The subspace Jacobian $\mathbf{J}_{\text{sub}} \in \mathbb{R}^{D_{\text{output}} \times P}$ is approximated column-wise via Finite Difference (FD). The $i$-th column, corresponding to the perturbation along the basis vector $\mathbf{w}_i$ of $\mathbf{W}$, is computed as:

$$\mathbf{J}_{\text{sub}} \cdot [\mathbf{w}_i] = \frac{G(\mathbf{z} + \epsilon \mathbf{w}_i) - G(\mathbf{z})}{\epsilon} \tag{14}$$

where $\epsilon$ is a sufficiently small perturbation radius. The resulting matrix has dimensions $\mathbf{J}_{\text{sub}} \in \mathbb{R}^{D_{\text{output}} \times P}$.

### B.1.2. METRIC TENSOR $\mathbf{A}$ AND EIGENDECOMPOSITION

The local geometry of the manifold is encapsulated by the metric tensor $\mathbf{A}$, defined as the product of the transpose of the subspace Jacobian and the Jacobian itself:

$$\mathbf{A} = \mathbf{J}_{\text{sub}}^{\text{T}} \mathbf{J}_{\text{sub}} \in \mathbb{R}^{P \times P} \tag{15}$$

This matrix represents the first fundamental form of the manifold restricted to the subspace $\mathbf{W}$. We perform eigendecomposition on this symmetric matrix $\mathbf{A}$:

$$\mathbf{A} = \mathbf{V} \mathbf{\Lambda} \mathbf{V}^{\text{T}} \tag{16}$$

where $\mathbf{\Lambda} = \text{diag}(\lambda_1, \lambda_2, \ldots, \lambda_P)$ contains the eigenvalues sorted in descending order, and $\mathbf{V} = [\mathbf{v}_1, \mathbf{v}_2, \ldots, \mathbf{v}_P]$ contains the corresponding eigenvectors.

### B.2. Geometric Descriptors

We isolate distinct geometric properties from the spectral components of the metric tensor.

### B.2.1. LOCAL SCALING $\psi_{\boldsymbol{\omega}}$

Local Scaling measures the **change in volume** of an infinitesimal region around $\mathbf{z}$ as it is mapped to the data space, relative to the $\mathbf{W}$ subspace. It is derived from the singular values $\sigma_i = \sqrt{\lambda_i}$:

$$\psi_{\boldsymbol{\omega}}(\mathbf{z}) = \sum_{i=1}^{P} \log(\sigma_i) \cdot \mathbf{1}_{\{\sigma_i > 0\}} \tag{17}$$

Expressed using eigenvalues $\lambda_i$, this quantifies the information capacity:

$$\psi_{\boldsymbol{\omega}}(\mathbf{z}) = \frac{1}{2} \sum_{i=1}^{P} \log(\lambda_i) \cdot \mathbf{1}_{\{\lambda_i > 0\}} \tag{18}$$

### B.2.2. PRINCIPAL DIRECTION $\mathbf{V}_1$

The eigenvector corresponding to the largest eigenvalue $\lambda_{\max}$ represents the **principal direction** of maximal change in the latent subspace:

$$\mathbf{V}_1(\mathbf{z}) = \mathbf{v}_{\max} \quad \text{such that } \mathbf{A}\mathbf{v}_{\max} = \lambda_{\max}\mathbf{v}_{\max} \tag{19}$$

This vector points in the direction where the generator is most sensitive to perturbations.

### B.2.3. LOCAL COMPLEXITY $\delta$

Local Complexity approximates the **un-smoothness or curvature** of the generative manifold in terms of the second-order change in the principal direction $\mathbf{V}_1$. It is computed as the averaged change of $\mathbf{V}_1$ over a neighborhood $\mathcal{N}_\epsilon(\mathbf{z})$:

$$\delta(\mathbf{z}) = \mathbb{E}_{\mathbf{z}' \sim \mathcal{N}_\epsilon(\mathbf{z})} \left[ \frac{\|\mathbf{V}_1(\mathbf{z}) - \mathbf{V}_1(\mathbf{z}')\|_2}{\|\mathbf{z} - \mathbf{z}'\|_2} \right] \tag{20}$$

where $\mathbf{z}'$ is a neighboring point such that $\|\mathbf{z} - \mathbf{z}'\|_2 = \epsilon$. High $\delta$ indicates that the principal axis of change rotates rapidly, implying a highly curved and unstable manifold surface.

## B.3. Principal Direction Projection ($\mathbf{P}_1$)

To link the abstract geometric properties of the latent space to the semantic content of the generated images, we define the Principal Direction Projection $\mathbf{P}_1$.

### B.3.1. DEFINITION OF $\mathbf{P}_1$

$\mathbf{P}_1$ is the projection of the latent space's principal change axis $\mathbf{V}_1$ onto the data space $\mathbf{x}$, computed via the Jacobian-Vector Product (JVP):

$$\mathbf{P}_1 = \mathbf{J}_{\text{sub}}\mathbf{V}_1 \in \mathbb{R}^{D_{\text{output}}} \tag{21}$$

### B.3.2. MATHEMATICAL ESSENCE (DIFFERENTIAL RATE)

The $\mathbf{P}_1$ vector represents the **instantaneous rate of change** of the image $\mathbf{x}$ when the latent code $\mathbf{z}$ is perturbed infinitesimally along the principal direction $\mathbf{V}_1$.

By the definition of the differential, where $\mathbf{J} = \frac{\partial G}{\partial \mathbf{z}}$:

$$\mathbf{P}_1 \approx \frac{dG}{d\mathbf{z}} \cdot \mathbf{V}_1 \tag{22}$$

Thus, $\mathbf{P}_1$ is a "change image" that visualizes the content being encoded by the local geometric primary axis $\mathbf{V}_1$.

### B.3.3. DERIVATION VIA TAYLOR EXPANSION

We can rigorously derive this approximation using the first-order Taylor expansion of the generator $G$ around $\mathbf{z}$. Let $\Delta\mathbf{z} = \epsilon\mathbf{V}_1$ be a perturbation along the principal unit vector. The generated output at the perturbed point is:

$$G(\mathbf{z} + \Delta\mathbf{z}) = G(\mathbf{z}) + \mathbf{J} \cdot \Delta\mathbf{z} + O(\|\Delta\mathbf{z}\|^2) \tag{23}$$

Substituting $\Delta\mathbf{z}$:

$$G(\mathbf{z} + \epsilon\mathbf{V}_1) - G(\mathbf{z}) \approx \epsilon\mathbf{J}\mathbf{V}_1 \tag{24}$$

Rearranging for the rate of change:

$$\frac{G(\mathbf{z} + \epsilon\mathbf{V}_1) - G(\mathbf{z})}{\epsilon} \approx \mathbf{J}\mathbf{V}_1 = \mathbf{P}_1 \tag{25}$$

Taking the limit as $\epsilon \to 0$ confirms that $\mathbf{P}_1$ is the directional derivative of the image generation function along the axis of maximum sensitivity.

## B.4. Projected High-Frequency Energy (PHFE)

PHFE quantifies the frequency content of the change image $\mathbf{P}_1$, serving as a critical diagnostic tool for geometric efficiency.

### B.4.1. MATHEMATICAL DEFINITION OF PHFE

PHFE is defined as the Mean Absolute Value (MAV) or Variance of the Laplacian response across the entire image dimension $D_{\text{output}}$. Using the MAV definition:

$$\text{PHFE} = \frac{1}{D_{\text{output}}} \sum_{j=1}^{D_{\text{output}}} |(\nabla^2 \mathbf{P}_1)_j| \tag{26}$$

Alternatively, using the Variance definition:

$$\text{PHFE} = \text{Var}(\mathbf{P}_1^{\text{laplace}}) = \text{Var}\left(\nabla^2 (\mathbf{J}_{\text{sub}} \mathbf{V}_1)\right) \tag{27}$$

### B.4.2. DIAGNOSTIC ROLE AND THEORETICAL IMPLICATIONS

The comparison between *Local Complexity* ($\delta$) and PHFE provides a quantitative basis for diagnosing Geometric Resource Misallocation:

- $\delta$ measures the **instability** (rotational speed of $\mathbf{V}_1$).

- PHFE measures the **content complexity** (high-frequency details) carried by $\mathbf{V}_1$.

**Interpretation:** If $\delta$ is high but PHFE is low, the geometric instability is not generating complex image details but is rather encoding non-semantic or low-frequency manifold twists. This state represents the **"Geometric Decoupling"**: to achieve local encoding efficiency, the LDM is forced to drive LC to extreme absolute values, thereby sacrificing geometric stability. This misallocation identifies "Geometric Hotspots" as the structural root of semantic jumps and interpolation failure.

### B.5. Theoretical Foundation of Spectral Isolation

We formally ground the Spectral Isolation Score (SIS) in Matrix Perturbation Theory, specifically the Davis-Kahan $\sin\Theta$ theorem (Davis & Kahan, 1970), to explain why SIS serves as a proxy for the local spectral gap and manifold rigidity.

Let $\mathbf{A} = \mathbf{J}^T\mathbf{J}$ be the local metric tensor at latent code $\mathbf{z}$, with eigenvalues $\lambda_1 > \lambda_2 \geq \cdots \geq \lambda_P$ and corresponding eigenvectors $\{\mathbf{v}'_1, \ldots, \mathbf{v}'_P\}$. Consider a perturbed point $\mathbf{z}' = \mathbf{z} + \epsilon$, which induces a perturbation matrix $\mathbf{E}$ such that the new metric tensor is $\tilde{\mathbf{A}} = \mathbf{A} + \mathbf{E}$. Let $\tilde{\mathbf{v}}_1$ (denoted as $\mathbf{V}_1$ in our experiments) be the perturbed principal eigenvector.

**First-Order Perturbation Expansion.** Assuming the perturbation $\|\mathbf{E}\|$ is small and the principal eigenvalue $\lambda_1$ is distinct (non-degenerate), we can approximate the perturbed eigenvector $\tilde{\mathbf{v}}_1$ as a linear combination of the original basis vectors using first-order perturbation theory:

$$\tilde{\mathbf{v}}_1 \approx \mathbf{v}_1 + \sum_{k=2}^{P} \frac{\mathbf{v}_k^T \mathbf{E} \mathbf{v}'_1}{\lambda_1 - \lambda_k} \mathbf{v}'_k \tag{28}$$

Here, the term $\mathbf{v}_k^T \mathbf{E} \mathbf{v}_1$ represents the projection of the perturbation noise onto the $k$-th axis, and $\Delta_k = \lambda_1 - \lambda_k$ represents the **spectral gap**.

**Derivation of SIS.** The cosine similarity terms in our SIS definition directly correspond to the coefficients in this expansion:

$$\text{CosSim}(\tilde{\mathbf{v}}_1, \mathbf{v}'_1) \approx 1 \quad \text{(Principal Stability)} \tag{29}$$

$$\text{CosSim}(\tilde{\mathbf{v}}_1, \mathbf{v}'_k) \approx \frac{|\mathbf{v}_k^T \mathbf{E} \mathbf{v}_1|}{\lambda_1 - \lambda_k} \quad \text{for } k > 1 \quad \text{(Leakage)} \tag{30}$$

Substituting these into the definition of SIS:

$$\text{SIS} = \frac{\text{CosSim}(\tilde{\mathbf{v}}_1, \mathbf{v}'_1)}{\sum_{k=2}^{P} \text{CosSim}(\tilde{\mathbf{v}}_1, \mathbf{v}'_k)} \approx \frac{1}{\sum_{k=2}^{P} \frac{|\mathbf{v}_k^T \mathbf{E} \mathbf{v}_1|}{\lambda_1 - \lambda_k}} \tag{31}$$

**Physical Interpretation.** The SIS is inversely proportional to the sum of leakage coefficients.

- **Low Spectral Gap** ($\lambda_1 \approx \lambda_2$): The denominator explodes as $\lambda_1 - \lambda_2 \to 0$. The principal direction is unstable and easily rotates into secondary dimensions. This results in a **Low SIS**, indicating a flexible or entangled manifold (typical of Normal generation).

- **High Spectral Gap** ($\lambda_1 \gg \lambda_2$): The denominator vanishes as $\lambda_1 - \lambda_k$ becomes large. The principal direction is "locked" by the large energy difference. This results in a **High SIS**, indicating a rigid, isolated manifold.

Thus, the pathological increase in SIS observed in OOD samples (SIS ↑) mathematically proves a widening of the local spectral gap. The model isolates the principal axis $\mathbf{V}_1$ from the rest of the spectrum to satisfy the semantic conflict, resulting in the "Tunnel Vision" geometry where the manifold loses its multidimensional plasticity.

## C. The Correlation Gap under Different prompts.

In table 10 and table 11 shown, we give the Spearman correlations between geometric measures and encoded detail with different prompts based on Stable Diffusion 3.5 and Flux.1. The significant drop in $\rho(\mathrm{LC}, \mathrm{PHFE})$, $\rho(\mathrm{LS}, \mathrm{PHFE})$ and $\rho(\mathrm{LS}, \mathrm{LC})$ for OOD samples quantifies the geometric decoupling.

*Table 11.* Spearman correlations between geometric measures and encoded detail with different prompts on Flux.1. The significant drop in $\rho(\mathrm{LC}, \mathrm{PHFE})$, $\rho(\mathrm{LS}, \mathrm{PHFE})$ and $\rho(\mathrm{LS}, \mathrm{LC})$ for OOD samples quantifies the geometric decoupling. Normal: Normal prompts, OOD: out-of-distribution prompts

*(a)* Rabbit vs Rabbit with deer antlers

|  | $\rho(\mathrm{LC}, \mathrm{PHFE})$ | $\rho(\mathrm{LS}, \mathrm{PHFE})$ | $\rho(\mathrm{LS}, \mathrm{LC})$ |
|---|---|---|---|
| Normal | 0.524 | 0.670 | 0.725 |
| OOD | 0.381 | 0.664 | 0.527 |

*(b)* Fish vs Fish with legs

|  | $\rho(\mathrm{LC}, \mathrm{PHFE})$ | $\rho(\mathrm{LS}, \mathrm{PHFE})$ | $\rho(\mathrm{LS}, \mathrm{LC})$ |
|---|---|---|---|
| Normal | 0.540 | 0.784 | 0.675 |
| OOD | 0.334 | 0.756 | 0.505 |

*(c)* Penguin vs Flying Penguin

|  | $\rho(\mathrm{LC}, \mathrm{PHFE})$ | $\rho(\mathrm{LS}, \mathrm{PHFE})$ | $\rho(\mathrm{LS}, \mathrm{LC})$ |
|---|---|---|---|
| Normal | 0.473 | 0.737 | 0.624 |
| OOD | 0.183 | 0.699 | 0.430 |

## D. Extended Statistical and Geometric Validations

In this section, we provide extended empirical results to validate the statistical robustness, dimensional dominance, and structural stability of the geometric decoupling phenomenon observed under Out-of-Distribution (OOD) stress.

### D.1. Dominance of the Principal Eigenvector

Our geometric analysis explicitly focuses on the principal eigenvector ($\mathbf{V}_1$) of the local Jacobian. We theoretically and empirically justify this focus through two key factors:

1. **Empirical Dominance:** The local Jacobian spectrum in LDMs is highly heavy-tailed. To evaluate this dominance, we calculated the cumulative eigenvalue ratio for the top-$k$ modes. As shown in Table 12, the principal eigenvalue ($\lambda_1$) alone accounts for 57.8%–59.9% of the total explained variance within the local random subspace. It is the absolute dominant direction of semantic variation.

2. **Theoretical Worst-Case Instability:** Geometrically, $\mathbf{V}_1$ represents the direction of maximum vulnerability (the steepest gradient of semantic change). By tracking the rotation and decoupling of this specific extreme direction, we precisely isolate the structural root cause of discontinuous semantic jumps and manifold brittleness.

*Table 12.* Cumulative eigenvalue ratio (explained variance) for the top-$k$ Jacobian modes.

| Label | Top-1 ($\lambda_1$) | Top-2 | Top-3 | Top-4 | Top-5 |
|---|---|---|---|---|---|
| ID | $0.599 \pm 0.136$ | $0.738 \pm 0.092$ | $0.799 \pm 0.076$ | $0.834 \pm 0.065$ | $0.858 \pm 0.058$ |
| OOD | $0.578 \pm 0.130$ | $0.722 \pm 0.091$ | $0.786 \pm 0.075$ | $0.823 \pm 0.065$ | $0.847 \pm 0.058$ |

## D.2. Statistical Robustness and Scaling Analysis

A core pillar of our claims is the absolute drop in correlation ($\rho$) between Local Complexity (LC) and Perceptible High-Frequency Energy (PHFE). To robustly verify the meaningfulness of these absolute values and ensure statistical significance, we scaled our evaluation from the initial 500 independent manifold neighborhoods up to 1,000 and 2,000 samples.

As demonstrated in Table 13, the results are definitive and scale-invariant. Regardless of sample size, the ID correlation remains stable at $\sim 0.41$ (proving that geometric curvature actively encodes image details), while it plummets to $\sim 0.09$ under OOD stress (degrading to pure statistical noise). Crucially, the 95% Confidence Intervals (CIs) for the ID and OOD regimes strictly never overlap. This precipitous absolute drop confirms the physical reality of functional "Geometric Decoupling" with extreme statistical robustness ($p < 0.05$).

*Table 13.* Statistical scaling of $\rho(\text{LC}, \text{PHFE})$ across $N = 500, 1000$, and 2000 sample sizes.

| Sample Size ($N$) | Regime | Metric | Spearman $\rho$ | 95% CI |
|---|---|---|---|---|
| | ID | LC vs PHFE@1 | 0.413 | $[0.347, 0.476]$ |
| | ID | LC vs PHFE@2 | 0.431 | $[0.365, 0.492]$ |
| | ID | LC vs PHFE@3 | 0.442 | $[0.376, 0.503]$ |
| 500 | | | | |
| | OOD | LC vs PHFE@1 | 0.082 | $[0.030, 0.136]$ |
| | OOD | LC vs PHFE@2 | 0.096 | $[0.041, 0.149]$ |
| | OOD | LC vs PHFE@3 | 0.108 | $[0.051, 0.161]$ |
| | ID | LC vs PHFE@1 | 0.414 | $[0.367, 0.458]$ |
| | ID | LC vs PHFE@2 | 0.437 | $[0.391, 0.480]$ |
| | ID | LC vs PHFE@3 | 0.451 | $[0.405, 0.493]$ |
| 1000 | | | | |
| | OOD | LC vs PHFE@1 | 0.092 | $[0.052, 0.132]$ |
| | OOD | LC vs PHFE@2 | 0.104 | $[0.062, 0.145]$ |
| | OOD | LC vs PHFE@3 | 0.117 | $[0.073, 0.158]$ |
| | ID | LC vs PHFE@1 | 0.414 | $[0.381, 0.446]$ |
| | ID | LC vs PHFE@2 | 0.433 | $[0.410, 0.474]$ |
| | ID | LC vs PHFE@3 | 0.456 | $[0.412, 0.497]$ |
| 2000 | | | | |
| | OOD | LC vs PHFE@1 | 0.099 | $[0.060, 0.138]$ |
| | OOD | LC vs PHFE@2 | 0.106 | $[0.075, 0.143]$ |
| | OOD | LC vs PHFE@3 | 0.118 | $[0.078, 0.156]$ |

## D.3. Stability of Principal Directions Across Regimes

To investigate the exact structural nature of the generative breakdown, we assessed whether the principal geometric axis is randomized across samples. We computed the sign-invariant absolute cosine similarity ($|\langle \mathbf{V}_1^{\text{ID}}, \mathbf{V}_1^{\text{OOD}} \rangle|$) for paired samples (*i.e.* transitioning from Normal to OOD using the exact same base prompt and random seed).

The results (Table 14) reveal that the principal directions remain remarkably stable, yielding a high mean similarity of 0.731 (median: 0.746). In an extremely high-dimensional latent space, this massive structural alignment proves that OOD stress

*Table 14.* Absolute cosine similarity of the principal axis ($\mathbf{V}_1$) between ID and OOD pairs.

| Group | $N$ | Mean | Std Dev | Median |
|---|---|---|---|---|
| ID vs OOD | 1000 | 0.731 | 0.139 | 0.746 |

does not randomize or destroy the principal geometric axis. Instead, the failure is purely *functional*: traversing this stable $\mathbf{V}_1$ axis in the Normal regime successfully encodes visual details (high PHFE), but traversing the exact same axis under OOD stress induces extreme structural curvature without generating meaningful image content.

## E. OOD (Out-of-Distribution) Prompt Construction

We construct an OOD prompt as a photorealistic description that combines a subject drawn from the COCO object set (Lin et al., 2014) with at least one structurally incompatible constraint of a fixed type (*e.g.*, physics violation, material–behaviour mismatch, semantic–function chimera, or scale/topology conflict). In contrast, a normal (non-OOD) prompt is physically realizable and semantically self-consistent under the same photorealistic setting. To control confounders, we construct paired prompts in which the normal and OOD versions share the same subject, composition, and lighting, and differ only in the contradiction clause. We further control prompt lengths by keeping the paired prompts approximately equal in length and use minimal, simple backgrounds to highlight the contradiction. The example prompts are given in Table 15.

*Table 15.* Example of normal and OOD prompts.

| Prompt Type | Prompt |
|---|---|
| OOD | A chicken with teeth, close-up of its beak showing small sharp teeth like a reptile, biological anomaly, scientific illustration style, high detail |
| Normal | A domestic chicken with a smooth, toothless beak, close-up portrait, natural farm setting, soft daylight, photorealistic, high detail |
| OOD | A fish with four muscular legs walking on sand in a desert, resembling a salamander but with fish scales and fins, speculative biology concept, photorealistic |
| Normal | A freshwater fish swimming underwater in a river, silver scales, flowing fins, natural aquatic environment with plants, sunlight filtering through water, photorealistic |
| OOD | A penguin flying high in the sky with large outstretched wings, clear view of wing feathers, snowy mountains below, physically incorrect avian behavior for a penguin, photorealistic, high detail |
| Normal | A penguin swimming underwater with flippers extended, bubbles trailing behind, icy ocean environment, natural behavior, photorealistic, crisp daylight filtering through water, high detail |

## F. Heat-map Visualization Methodology

We compute the pixel-wise Frobenius norm of the Jacobian $\mathbf{J} = \partial G(\mathbf{z})/\partial \mathbf{z}$ by aggregating the squared gradients of the output features with respect to the input latent tensor. To diagnose the functional utility of this sensitivity, we visualize the spatial structure of the principal change vector $\mathbf{P}_1 = \mathbf{J}\mathbf{V}_1$. We apply a discrete Laplacian operator $\nabla^2$ to $\mathbf{P}_1$ and visualize the magnitude $|\nabla^2 \mathbf{P}_1|$. All heatmaps are upsampled to the image resolution and normalized to highlight relative intensity differences between structural boundaries and background noise.

### F.1. Additional results of Heat-map Visualization

This appendix provides additional qualitative examples to further validate the "Geometric Decoupling" phenomenon discussed in the main text. We present an extended set of visualizations comparing the generated OOD samples with their corresponding Local Complexity Maps (LC-Map) and Projected High-Frequency Energy Maps (PHFE-Map). The LC-Map visualizes the spatial distribution of the manifold's sensitivity ($\|\nabla_{\mathbf{z}}\mathbf{x}\|_F$), while the PHFE-Map indicates the functional utility of this sensitivity.

| $k$ | Normal | OOD | (OOD-Normal) | (OOD/Normal) |
|---|---|---|---|---|
| 0 | 0.2519±0.0114 | 0.3013±0.0107 | 0.0324±0.0180 | 1.100±0.057 |
| 1 | 0.2511±0.0114 | 0.2930±0.0103 | 0.0297±0.0193 | 1.093±0.062 |
| 2 | 0.2519±0.0125 | 0.3015±0.0110 | 0.0341±0.0194 | 1.103±0.061 |
| 3 | 0.2539±0.0115 | 0.3047±0.0123 | 0.0531±0.0186 | 1.163±0.061 |
| 4 | 0.2682±0.0119 | 0.3028±0.0111 | 0.0241±0.0203 | 1.072±0.061 |
| 5 | 0.2611±0.0110 | 0.3154±0.0120 | 0.0490±0.0207 | 1.147±0.066 |
| 6 | 0.2587±0.0113 | 0.3099±0.0115 | 0.0515±0.0169 | 1.155±0.055 |
| 7 | 0.2546±0.0115 | 0.3112±0.0114 | 0.0539±0.0186 | 1.168±0.063 |
| 8 | 0.2598±0.0120 | 0.2998±0.0118 | 0.0429±0.0209 | 1.130±0.066 |
| 9 | 0.2679±0.0122 | 0.3062±0.0117 | 0.0213±0.0183 | 1.061±0.053 |
| 10 | 0.2554±0.0104 | 0.3059±0.0118 | 0.0566±0.0191 | 1.176±0.063 |
| 11 | 0.2559±0.0110 | 0.2932±0.0115 | 0.0323±0.0187 | 1.101±0.061 |
| 12 | 0.2569±0.0119 | 0.3114±0.0127 | 0.0667±0.0209 | 1.202±0.069 |
| 13 | 0.2729±0.0131 | 0.2995±0.0112 | 0.0331±0.0162 | 1.099±0.051 |
| 14 | 0.2633±0.0114 | 0.3018±0.0118 | 0.0406±0.0199 | 1.122±0.063 |
| 15 | 0.2494±0.0108 | 0.3078±0.0124 | 0.0690±0.0181 | 1.226±0.066 |
| 16 | 0.2545±0.0115 | 0.3119±0.0128 | 0.0562±0.0191 | 1.172±0.062 |
| 17 | 0.2756±0.0113 | 0.3005±0.0115 | 0.0187±0.0208 | 1.056±0.063 |
| 18 | 0.2499±0.0102 | 0.2980±0.0110 | 0.0592±0.0204 | 1.195±0.073 |
| 19 | 0.2579±0.0123 | 0.2968±0.0106 | 0.0384±0.0202 | 1.118±0.065 |
| Across $k$ (mean) | 0.2585 | 0.3036 | 0.0431 | 1.133 |

*Table 16.* Monte-Carlo estimates of the latent trajectory increments at each interpolation step $k$ for Normal versus OOD conditions. We report the difference $\Delta$ (OOD - Normal) and the ratio (OOD/Normal) as mean $\pm$ standard deviation across Monte Carlo resamples ($n_{\mathrm{mc}} = 800$, sample fraction 0.8, sample size 100, with replacement). The consistently higher increments for OOD samples across all steps indicate a globally expanded and less efficient traversal path.

As shown in Figures 2 and 3, these supplementary results, generated by Stable Diffusion 3.5 and Flux.1, are consistent with our primary conclusions. We observe that "Geometric Hotspots", regions of extreme curvature highlighted in red on the LC-Maps, systematically align with semantic anomalies, such as the unnatural teeth of a chicken, the melting structure of a chair, or the wings of a penguin. Crucially, these high-curvature regions often exhibit a disconnect from the functional high-frequency details shown in the PHFE-Maps, reinforcing that under OOD conditions, geometric resources are misallocated to resolving structural conflicts rather than encoding perceptible details.

## G. Interpolation Trajectory

This appendix provides the granular, step-by-step breakdown of the latent trajectory increments observed during the spherical linear interpolation (slerp) experiments. Table 16 details the mean displacement magnitude ($\Delta_k$) at each discretization step $k$ along the interpolation path $\gamma(t)$ for $t \in [0, 1]$.

The data demonstrates a **systematic geometric expansion** under Out-of-Distribution (OOD) conditions. Across all interpolation steps $k = 0 \ldots 19$, the trajectory increments for OOD samples are consistently larger than those for Normal (In-Distribution) samples, with a mean ratio of approximately 1.133 (representing a $\sim 13.3\%$ increase in local path length). This uniformity indicates that the "Geometric Decoupling", where the manifold becomes locally stretched and inefficient, is a global property of the OOD latent space traversal, rather than an artifact isolated to specific segments of the path.

## H. Different $k$ for Top$k$-HF (HF concentration)

We report Top5/Top10-HF in the main text as they are the most sensitive probes of sparsity/concentration in the strongest high-frequency responses. Larger thresholds (Top15/Top20) progressively include moderate-magnitude regions and thus dilute the contrast as Table 17 shows.

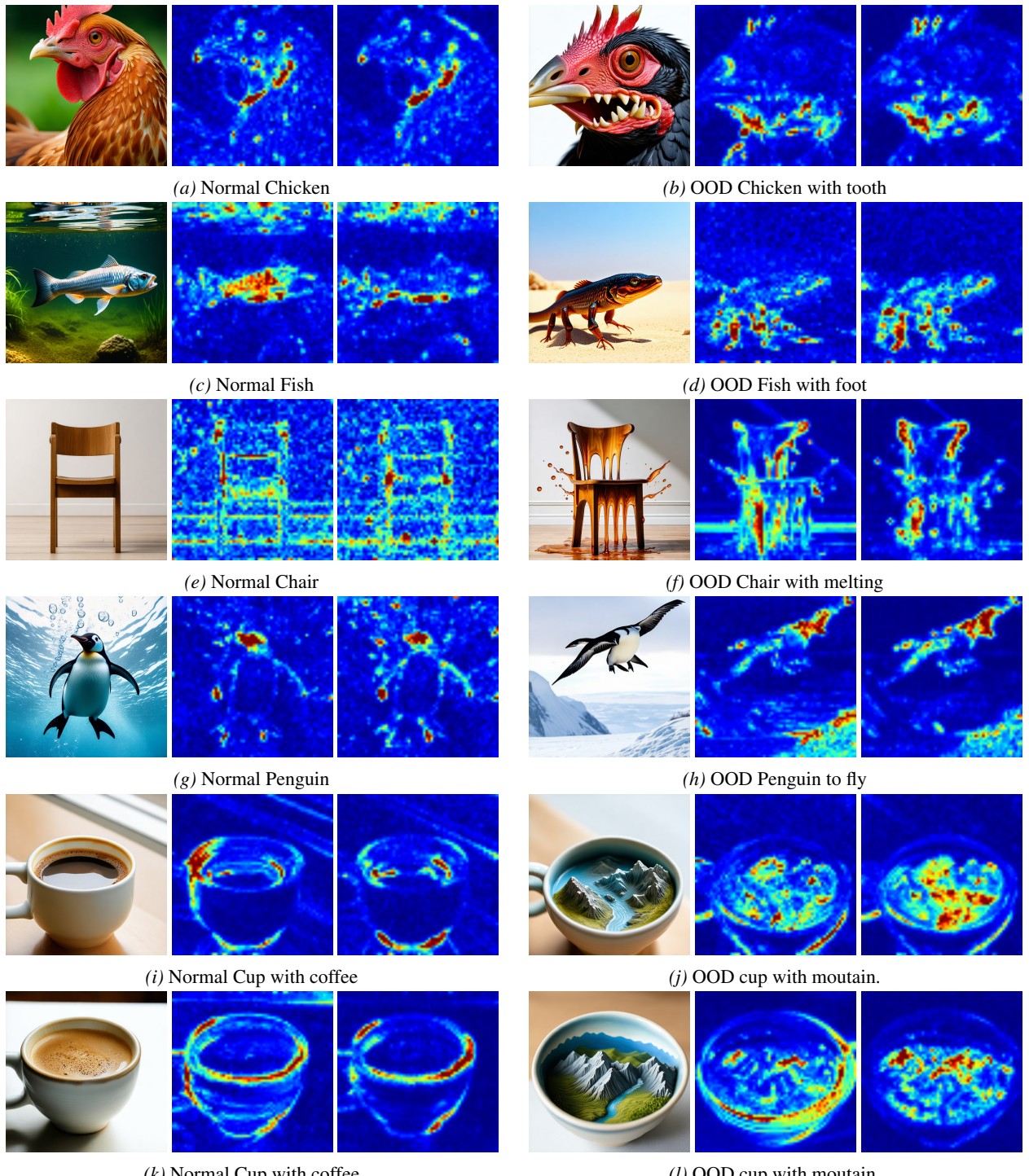

*(a)* Normal Chicken

*(b)* OOD Chicken with tooth

*(c)* Normal Fish

*(d)* OOD Fish with foot

*(e)* Normal Chair

*(f)* OOD Chair with melting

*(g)* Normal Penguin

*(h)* OOD Penguin to fly

*(i)* Normal Cup with coffee

*(j)* OOD cup with moutain.

*(k)* Normal Cup with coffee

*(l)* OOD cup with moutain.

*Figure 2.* **Qualitative Visualization of Geometric Decoupling.** We display OOD samples alongside their Local Complexity Maps (LC-Map) and Projected High-Frequency Energy Maps (PHFE-Map) for Stable Diffusion 3.5. In each subfigure, from left to right: the Generated Image, the LC-Map, and the PHFE-Map. Red regions in the LC-Map denote "Geometric Hotspots" of extreme curvature. These hotspots align precisely with semantic anomalies, for example, the teeth of a chicken, the liquefaction of a chair, or the wings of a penguin. This spatial correspondence confirms that the model allocates its maximum geometric complexity to resolving semantic conflicts, often decoupling from the actual high-frequency detail (PHFE), thereby illustrating the misallocation of geometric resources.

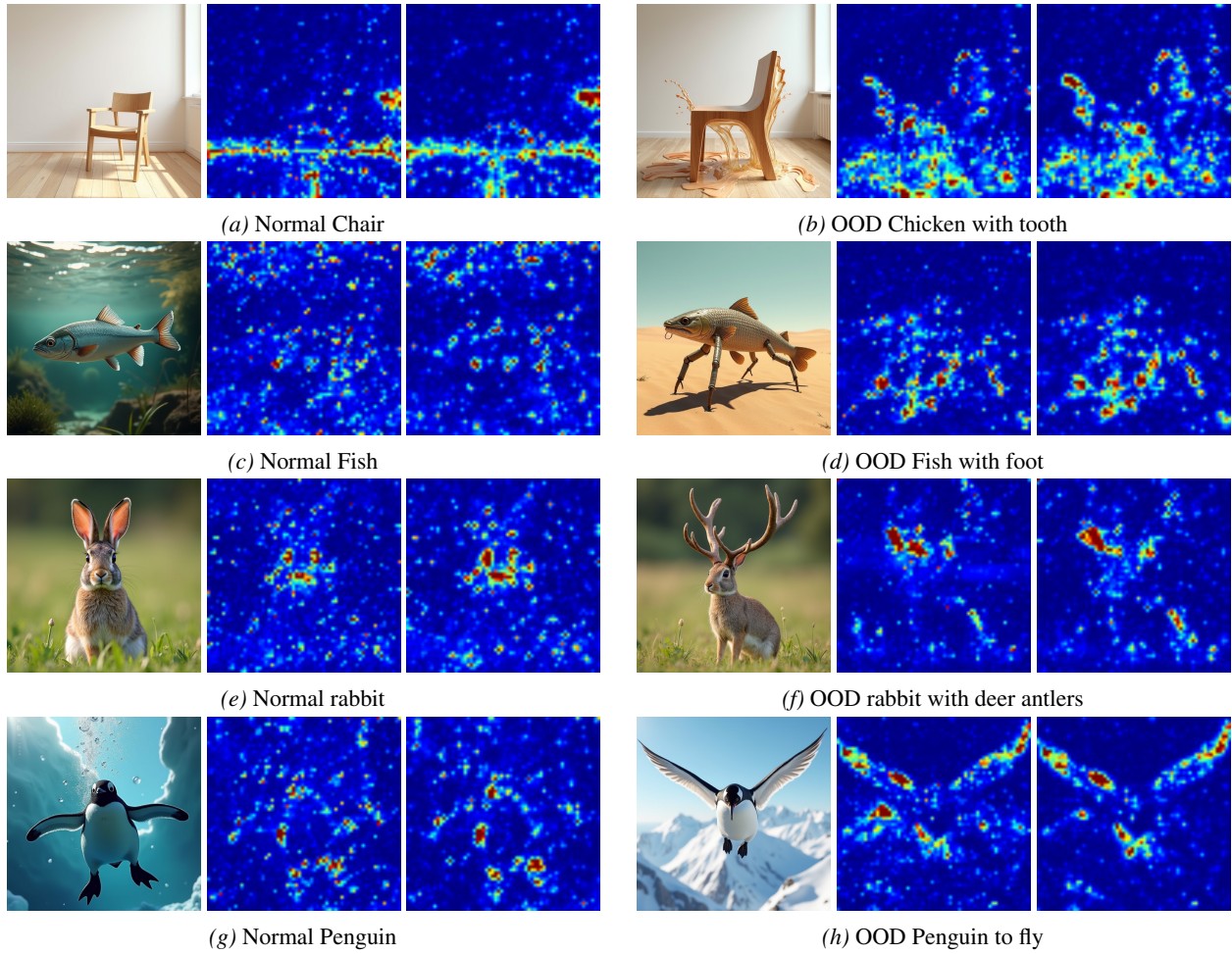

*(a)* Normal Chair

*(b)* OOD Chicken with tooth

*(c)* Normal Fish

*(d)* OOD Fish with foot

*(e)* Normal rabbit

*(f)* OOD rabbit with deer antlers

*(g)* Normal Penguin

*(h)* OOD Penguin to fly

*Figure 3.* **Qualitative Visualization of Geometric Decoupling.** We display OOD samples alongside their Local Complexity Maps (LC-Map) and Projected High-Frequency Energy Maps (PHFE-Map) for Flux.1. In each subfigure, from left to right: the Generated Image, the LC-Map, and the PHFE-Map. Red regions in the LC-Map denote "Geometric Hotspots" of extreme curvature. These hotspots align precisely with semantic anomalies, for example, the teeth of a chicken, the liquefaction of a chair, or the wings of a penguin. This spatial correspondence confirms that the model allocates its maximum geometric complexity to resolving semantic conflicts, often decoupling from the actual high-frequency detail (PHFE), thereby illustrating the misallocation of geometric resources.

*Table 17.* High-frequency concentration statistics (median) across different Top$k$ thresholds. Lower values indicate more spatially diffuse (noise-like) high-frequency patterns.

|  | Top5-HF | Top10-HF | Top15-HF | Top20-HF |
|---|---|---|---|---|
| ID | 0.530 | 0.691 | 0.765 | 0.813 |
| OOD | 0.493 | 0.664 | 0.757 | 0.815 |

## I. Limitations and Future work

**Limitations** Our study faces two primary limitations. First, the matrix-free approximation of the high-dimensional Jacobian is computationally intensive, restricting real-time application during inference. Second, and most critically, our geometric diagnosis is contingent on the **manifestation** of the OOD event. We observed that certain abstract OOD categories, particularly Physics Violations (*e.g.*"water flowing upwards"), are difficult to observe empirically. The model's learned physical priors are often robust enough to override the text prompt, generating a realistic (Normal) image despite the OOD instruction. In such cases where the hallucination fails to materialize, the latent geometry remains stable, preventing the observation of the "Geometric Decoupling". Thus, our metric serves as a detector of *realized* structural failure rather than latent semantic intent.

**Future Work.** These findings suggest a new direction for "Geometric-Aware" generative modeling. Future work should explore **Geometric Regularization** terms during training that penalize high Local Complexity variance, forcing the model to learn smoother transitions at semantic boundaries. Additionally, **Curvature-Adaptive Sampling** could be developed to dynamically adjust step sizes during inference.

