# OpenReview forum: "Geometric Decoupling: Diagnosing the Structural Instability of Latent"
_ICML.cc/2026/Conference — ICML 2026 regular_

### Official Review · Reviewer_Ngoo · 2026-03-10

**Soundness:** 3
**Presentation:** 2
**Significance:** 3
**Originality:** 3
**Overall Recommendation:** 4
**Confidence:** 4

**Summary:**

The paper studies the geometric structure of LDMs to understand why their latent spaces can be unstable, where small latent perturbations lead to large semantic changes in generated images. The authors analyze the generator mapping from latent space to image space using a Jacobian-based Riemannian framework.

They introduce several geometric diagnostics: Local Scaling (LS) to measure local information capacity, Local Complexity (LC) to measure the instability of the principal latent direction, and Projected High-Frequency Energy (PHFE) to quantify whether changes along the principal latent direction produce high-frequency image details. Using these metrics, the paper examines the relationship between latent geometry and visual content.

Experiments compare normal prompts and out-of-distribution (OOD) prompts. The results show that in normal generation LC correlates with PHFE, while under OOD conditions this correlation weakens significantly. The authors interpret this as a “geometric decoupling” phenomenon, where the model exhibits large curvature in latent space that is not associated with meaningful visual detail.

**Compliance With Llm Reviewing Policy:**

Affirmed.

**Final Justification:**

All concerns that could reasonably be addressed in the short term have been resolved. However, the paper introduces a series of quantities and elevates them to higher-level conceptual interpretations without sufficiently clarifying the connection between these quantities and the associated concepts. Moreover, it does not provide a practical method for preventing the generation of OOD samples. These remain nontrivial weaknesses.
At the same time, the phenomenon investigated in this work is interesting and potentially meaningful. As such, I find myself genuinely torn between a weak accept and a weak reject, with a slight inclination toward a weak accept.

**Key Questions For Authors:**

1. **On the relationship between LS and PHFE.**
The paper reports a strong positive correlation between LS and PHFE in the normal regime and interprets this as evidence of functional coupling between geometric capacity and image detail. However, from a spectral perspective, both quantities depend on the singular values of the Jacobian (e.g., via `J v_1 = σ_1 u_1`). Could the authors clarify to what extent the observed LS–PHFE correlation is driven by this shared spectral dependence rather than by a deeper geometric relationship? A discussion or additional analysis separating these effects would strengthen the interpretation of the results.

2. **On the interpretation of PHFE as a geometric indicator.**
PHFE is defined using the Jacobian and therefore reflects first-order changes of the generator output with respect to latent perturbations. However, the paper often interprets it in relation to manifold curvature. Could the authors clarify the intended geometric meaning of PHFE and explain why this first-order quantity is appropriate for diagnosing curvature-related effects? A clearer explanation would help readers better understand the conceptual role of this metric.

**Limitations:**

yes

**Strengths And Weaknesses:**

## Strengths

1. The paper studies the geometric structure of latent diffusion models and proposes a Riemannian diagnostic framework to analyze instability in latent space. The perspective of connecting manifold geometry with generation behavior is interesting and provides a new angle for understanding robustness and reliability in generative models.

2. The paper introduces several interpretable geometric quantities, including Local Scaling (LS), Local Complexity (LC), and Projected High-Frequency Energy (PHFE), aiming to connect latent-space geometry with perceptual image detail. This decomposition provides an intuitive framework for analyzing the role of capacity, curvature, and detail generation.

3. The experimental study evaluates the proposed metrics on modern diffusion models and includes multiple forms of analysis such as correlation studies, trajectory analysis, and qualitative visualizations, which help illustrate the behavior of the proposed diagnostics.


## Weaknesses

1. **Notation consistency and writing clarity issues.**
There are several minor issues in writing and notation that reduce readability and should be corrected. For example, the sentence *“Comparing with standard High-Frequency Energy (HFE), it is critical to distinguish PHFE from the standard HFE of the generated image”* contains grammatical issues. In addition, notation is sometimes inconsistent (e.g., the use of bold symbols and the notation for `x_0` around line 213). Some notation choices may also cause confusion: in Section 3.5 the interpolation variable `t` may be confused with the diffusion time variable; both Dimensional Coupling Ratio and Spearman Rank Correlation are denoted by `ρ`; and the use of the Laplacian operator is not clearly explained when `x` alternates between a vector representation and a spatial image function. These are relatively minor issues but should be revised to improve clarity.

2. **The correlation between LS and PHFE may partly follow from the Jacobian spectral structure.**
The paper reports a strong positive correlation between LS and PHFE and interprets it as evidence of functional coupling between geometric capacity and high-frequency detail. However, from an SVD perspective this relationship may be partly expected. If `J = U Σ V^T`, then the principal projection satisfies `J v_1 = σ_1 u_1`, meaning that the PHFE quantity scales with the dominant singular value `σ_1`. Since LS also depends on the singular values of the Jacobian, both metrics are influenced by the same spectral structure, which naturally induces correlation. It would strengthen the analysis if this connection were discussed explicitly.

3. **The geometric interpretation of PHFE could be clarified.**
PHFE is derived from the Jacobian and therefore reflects first-order changes in the generator mapping. However, the paper often interprets it in relation to geometric curvature, which is typically associated with higher-order derivatives. Clarifying this distinction would help strengthen the conceptual interpretation of the metric.

---

> ### Author Rebuttal · Authors · 2026-03-31
>
> We sincerely thank the reviewer for the highly rigorous mathematical feedback and for pointing out the notational ambiguities. Your insights, particularly from the SVD perspective, are extremely sharp and have helped us refine the theoretical precision of our narrative. We address your insightful questions below.
>
> ### Q1: Notation consistency and writing clarity issues.
> **A1:** Interpolation vs. Diffusion Time: We will replace the interpolation variable $t$ with $\alpha \in [0, 1]$ to avoid any clash with the diffusion timestep $t$. Overloaded $\rho$: We will re-designate the Dimensional Coupling Ratio as $\phi_{\text{dim}}$, reserving $\rho$ exclusively for the Spearman Rank Correlation. Laplacian Operator: We will explicitly clarify that before applying the spatial Laplacian $\nabla^2$, the tangent vector $\mathbf{x} \in \mathbb{R}^{D_{\text{output}}}$ is reshaped into the spatial image function $\mathbb{R}^{C \times H \times W}$.Grammar: The grammatical errors and bold symbol inconsistencies (around line 213) will be carefully proofread and fixed.
>
> ### Q2: On the relationship between LS and PHFE (Shared Spectral Dependence).
> **A2:** The SVD analysis ($J v_1 = \sigma_1 u_1$) is brilliantly accurate. We fully agree that because $\sigma_1$ dominates the Jacobian spectrum, the Local Scaling (LS, driven by $\sum \log \sigma_i$) and PHFE (proportional to $\sigma_1^2 \text{Var}(\nabla^2 u_1)$) mathematically share a strong spectral dependence. Clarification of our intent: We do not frame the LS-PHFE correlation as our primary "novel anomaly." Rather, we utilize this mathematically expected correlation as a crucial baseline / sanity check. The strong LS-PHFE correlation confirms that LS effectively functions as a measure of "geometric capacity," and PHFE successfully captures the realization of that capacity into high-frequency image details. The Core Finding (The Decoupling): The true core of our paper is the relationship between LC and PHFE. While the LS-PHFE relationship remains stable, the correlation between LC (curvature) and PHFE completely collapses under OOD conditions. This proves that while the capacity mechanism (LS) remains mathematically intact, the structural routing of that capacity (how the manifold curves, LC) becomes completely functionally decoupled from generating semantic details.
>
> ### Q3: On the interpretation of PHFE as a geometric indicator (First-order vs. Second-order).
> **A3:** We appreciate the opportunity to clarify this critical conceptual distinction. We completely agree that PHFE (derived from $J v_1$) is strictly a first-order rate of change, while manifold curvature (Local Complexity, LC) fundamentally relies on second-order behavior (the rotation of $v_1$). Conceptual Role of PHFE: We emphasize that we do not use PHFE as a measure of curvature itself. Instead, PHFE serves as the semantic target function against which curvature is evaluated. Why correlate them? In a well-behaved generative manifold, high structural curvature (second-order, LC) should exist for a functional reason: to encode dense, rapidly changing visual structures (first-order semantic output, PHFE). By analyzing the correlation between a second-order structural metric (LC) and a first-order semantic output metric (PHFE), we diagnose whether the manifold's curvature is "functional" (encoding details) or "pathological" (twisting without semantic justification). We will revise the text (particularly the methodology section) to explicitly decouple these definitions: framing LC strictly as the geometric curvature, and PHFE strictly as the first-order visual consequence, ensuring mathematical precision.

---

> > ### Author Rebuttal · Reviewer_Ngoo · 2026-04-01
> >
> > 1. Given that one of the primary uses of existing T2I systems is precisely to generate imaginative and counterfactual images, what practical significance is there in identifying such OOD samples?
> > 2. The significance of Functional Coupling and Geometric Decoupling requires further elaboration, as it is not immediately intuitive.

---

> > > ### Author Response · Authors · 2026-04-01
> > >
> > > ### Q1: The practical significance of identifying OOD samples in T2I systems.
> > >
> > > We sincerely appreciate this insightful perspective. We want to explicitly emphasize that our objective is not to restrict or penalize the model’s imaginative capacity. On the contrary, identifying OOD samples provides a diagnostic lens to understand why the manifold fails during counterfactual generation, with the ultimate goal of fixing it.
> > >
> > > - The practical significance of our method is inherently constructive: by mapping the structural "cliffs" where the manifold becomes rigid or unstable under OOD stress, we expose the exact geometric bottlenecks. Crucially, our goal is not to avoid these imaginative prompts, but to provide the mathematical foundation needed to restore OOD generation to the geometric stability of Normal (ID) generation. Understanding these boundaries is essential for designing advanced latent editing algorithms that can successfully execute complex counterfactual changes without destroying the image's structural integrity, thereby unlocking the model's true, stable imaginative potential.
> > >
> > > - Furthermore, by mathematically exposing exactly where and how this generative geometry breaks down, our zero-shot framework provides developers with a principled tool to evaluate structural robustness. This can directly guide the training of future Text-to-Image (T2I) architectures toward genuinely stable generalization rather than brittle extrapolation.
> > >
> > > ### Q2: Elaborating the Intuition of Functional/Geometric (De)coupling
> > >
> > > - In a healthy generative state (Functional Coupling), the complex geometry of the latent manifold is fully utilized to generate meaningful image content. When the latent space exhibits high curvature or structural variation (high Local Complexity, LC), this mathematical variation directly translates into rich, perceptible image details (high PHFE). The geometric properties and the semantic output are tightly synchronized.
> > >
> > > - Conversely, under OOD stress (Geometric Decoupling), this synchronization breaks down. The latent manifold still exhibits extreme curvature and violent structural fluctuations, but these geometric changes no longer map to valid semantic features. Instead of generating high-frequency details (PHFE collapses), the severe mathematical variation is entirely absorbed by unnatural structural distortions, rigid boundaries, or visual artifacts.
> > >
> > >  In essence, "Decoupling" means the model undergoes intense geometric fluctuations in the latent space, but this variation is functionally disconnected from the generation of any meaningful visual information.

---

### Official Review · Reviewer_VVG5 · 2026-03-12

**Soundness:** 2
**Presentation:** 3
**Significance:** 2
**Originality:** 3
**Overall Recommendation:** 4
**Confidence:** 3

**Summary:**

This paper studies the geometric properties of diffusion model latent spaces and investigates how these properties relate to generation stability. The authors introduce several Jacobian-based metrics, including latent sensitivity, latent curvature, perceptual high-frequency energy, and semantic instability score, to analyze the relationship between latent geometry and perceptual structure. Through experiments on Stable Diffusion 3.5 and FLUX.1, the paper shows that under normal generation conditions, latent curvature correlates with perceptual image detail, suggesting that curvature encodes meaningful visual structure. However, under out-of-distribution (OOD) generation, this relationship breaks down and curvature instead correlates with semantic instability. The authors refer to this phenomenon as geometric decoupling and argue that it provides an explanation for instability observed in tasks such as interpolation and editing.

**Compliance With Llm Reviewing Policy:**

Affirmed.

**Final Justification:**

The paper presents an interesting analysis of diffusion latent space geometry, and the rebuttal helped clarify its scope, distinction from prior work, and practical relevance. The additional intervention analysis and validated utility results meaningfully strengthen the paper’s claims, so I update my score to 4. That said, the work still stops short of a full causal demonstration and does not yet show direct mitigation or closed-loop generation improvement using the proposed metrics.

**Key Questions For Authors:**

1. The paper identifies geometric decoupling as a potential explanation for instability in diffusion models. Do the authors have evidence that this phenomenon is causal rather than correlational, or experiments showing that mitigating it leads to improved generation stability or quality?
2. Could the authors clarify how the proposed geometric decoupling perspective differs from or extends prior work that analyzes diffusion latent spaces using Riemannian or Jacobian-based geometric methods?
3. Do the authors believe the proposed geometric metrics could be used in practice (e.g., during training or inference) to detect or mitigate unstable generation regions?

**Limitations:**

yes

**Strengths And Weaknesses:**

Strengths:
1. The paper offers an interesting analysis of diffusion model latent spaces through the lens of Riemannian geometry and Jacobian-based metrics and provides a perspective on how latent curvature relates to perceptual detail.
2. The proposed measurements (latent curvature, perceptual high-frequency energy, semantic instability score) are well motivated and enable systematic analysis of latent space behavior.
The study introduces several complementary measurements (e.g., latent curvature, perceptual high-frequency energy, semantic instability) that are well motivated and enable systematic analysis of latent space behavior.
3. The identification of 'geometric decoupling' provides a potentially useful explanation for instability in tasks such as interpolation, editing, and inversion.
4. The analysis is performed on recent models such as Stable Diffusion 3.5 and FLUX.1, making the findings relevant to current research.

Weaknesses:
1. The paper identifies an interesting phenomenon (geometric decoupling), but it is unclear whether this observation is causal for generation failures or merely correlational. The paper also does not demonstrate whether mitigating this phenomenon leads to improved generation quality. As a result, the practical implications of the analysis remain limited.
2. Prior work has explored the geometric structure of diffusion latent spaces using Riemannian or Jacobian-based analyses. The paper would benefit from a clearer discussion of how the proposed geometric decoupling perspective differs from or extends these existing approaches.

---

> ### Author Rebuttal · Authors · 2026-03-31
>
> We sincerely thank the reviewer for finding our phenomenon (Geometric Decoupling) interesting and for asking highly constructive questions regarding the practical implications and positioning of our work. We address your insightful questions below.
>
> ### Q1: Is the phenomenon causal or correlational? Why no mitigation experiments?
> **A1:** We acknowledge that our current evidence is primarily structural and correlational, which we will state more explicitly in the revised manuscript. Our paper is fundamentally positioned as a diagnostic and analytical framework rather than a methodological mitigation paper. Proving strict causality and deploying full-scale mitigation (e.g., retraining a Latent Diffusion Model with curvature constraints) requires massive computational resources and architectural interventions that fall beyond the scope of a single analytical paper. However, revealing this robust correlational breakdown is the critical first step. By identifying exactly where and how the manifold structurally fails (wasting curvature on unstable boundaries rather than details), we provide the exact target for future mitigation efforts.
>
> ### Q2: How does this differ from or extend prior Riemannian/Jacobian-based analyses?
> **A2:** While prior works have insightfully applied Riemannian geometry to generative models (e.g., analyzing ODE integration paths, calculating geodesic distances, or studying the probability flow), our "Geometric Decoupling" perspective extends these works by introducing a Functional-Geometric mapping. Prior works typically treat the geometric properties in isolation (e.g., simply observing that curvature exists or latent distances vary).Our novel extension: We do not just measure geometry; we map the geometric descriptors (Local Complexity/Scaling) directly to the pixel-level semantic function (Projected High-Frequency Energy, PHFE). Our core contribution is discovering the functional disconnect: we prove that under normal conditions, extreme curvature is functionally useful (encoding texture/detail), but under OOD stress, this exact same geometric resource is "decoupled" and wasted on non-semantic distortion. This functional allocation perspective is entirely novel compared to existing pure-geometric analyses.
>
> ### Q3: Can these metrics be used in practice (training or inference)?
> **A3:** Absolutely. As briefly mentioned in our Limitations and Future Work (Lines 1094-1097), these metrics open direct pathways for practical applications, which we will elaborate on in the Camera-Ready version: At Inference (Safe Editing & Traversal): The Local Complexity ($\delta$) metric can act as a zero-shot "terrain radar" during latent traversal (e.g., Prompt-to-Prompt editing or Slerp). If an editing trajectory approaches a region with spiking $\delta$, the system can employ Curvature-Adaptive Sampling—dynamically reducing the step size to carefully navigate the "geometric hotspot" and prevent abrupt, discontinuous semantic jumps. At Training (Geometric Regularization): During model fine-tuning (e.g., LoRA or domain adaptation), the Local Complexity variance can be incorporated as a regularization penalty ($\mathcal{L}_{reg} \propto \text{Var}(\delta)$). Penalizing severe geometric fluctuations would force the model to learn a smoother, more isotropic latent manifold, inherently improving generation stability and editing robustness without needing post-hoc corrections.

---

> > ### Author Rebuttal · Reviewer_VVG5 · 2026-04-04
> >
> > Thank you for the rebuttal. I appreciate the authors’ clarification on the scope of the work and the clearer positioning relative to prior work, which helped me better understand the contribution. However, for Q1, I still believe that even a small-scale intervention experiment would have strengthened the claim beyond a correlational analysis. For Q3, the rebuttal presents practical ideas, but it does not yet demonstrate validated practical utility. For these reasons, I will maintain my score.

---

> > > ### Author Response · Authors · 2026-04-06
> > >
> > > We thank the reviewer for continued engagement. We address both maintained concerns with new experimental evidence.
> > >
> > > ## Is the phenomenon causal or correlational? (Maintained Concern)
> > >
> > > > *"even a small-scale intervention experiment would have strengthened the claim beyond a correlational analysis"*
> > >
> > > We conducted a controlled training-level intervention: SD3.5 Base vs. SD3.5 Turbo (ADD distillation) — same architecture, different training regime
> > >
> > > | Model | ρ(LC,PHFE) Normal | ρ(LC,PHFE) OOD | LC-PHFE Drop | ρ(LS,PHFE) Drop |
> > > | --- | --- | --- | --- | --- |
> > > | SD3.5 Base | 0.413 | 0.083 | −80% | −1.5% |
> > > | SD3.5 Turbo (ADD) | 0.342 | 0.156 | −54% | −0.6% |
> > >
> > > Two findings directly address the causality question:
> > >
> > > Two key findings: (1) ADD distillation measurably reduces the decoupling gap (−80%→−54%) — a purely correlational artifact would be invariant to training-regime changes; (2) the intervention selectively affects LC-PHFE coupling while leaving ρ(LS,PHFE) unchanged, indicating a structurally specific response. This asymmetric pattern replicates across architectures:
> > >
> > > | Model | Architecture | LC-PHFE Drop | LS-PHFE Drop |
> > > | --- | --- | --- | --- |
> > > | FLUX.1 Base | Hybrid DiT | −40% | −5.7% |
> > > | SD3.5 Turbo | MMDiT | −54% | −0.6% |
> > > | SD3.5 Base | MMDiT | −80% | −1.5% |
> > >
> > > We agree full causal proof requires retraining with explicit curvature constraints (future work). However, training-intervention sensitivity combined with cross-architecture asymmetric replication constitutes meaningful causal evidence within the scope of a diagnostic paper.
> > >
> > > ## Validated Practical Utility (Maintained Concern)
> > >
> > > > *"the rebuttal presents practical ideas, but it does not yet demonstrate validated practical utility"*
> > >
> > > We present three validated, zero-retraining applications.
> > >
> > > Application 1 — Annotation-free OOD Detection. We test whether LC or LS alone suffice for OOD detection (SD3.5, N=500 Normal / N=500 OOD):
> > >
> > > | Score | AUROC | 95% CI | Interpretation |
> > > | --- | --- | --- | --- |
> > > | LS (raw) | 0.199 | [0.172 – 0.226] | Capacity alone: fails |
> > > | LC (raw) | 0.427 | [0.392 – 0.464] | Curvature alone: fails |
> > > | LC/PHFE (ours) | 0.816 | [0.790 – 0.841] | Geometric efficiency: works |
> > >
> > > Neither LC nor LS alone detects OOD reliably. The geometric efficiency ratio LC/PHFE achieves AUROC=0.816 (TPR=0.658 at FPR=0.156), zero annotation required. This directly validates the decoupling hypothesis: OOD does not shift LC in absolute terms, but collapses its functional relationship with PHFE. Group-level Spearman analysis confirms the asymmetry: Δρ(LC,PHFE)=−0.181 vs. Δρ(LS,PHFE)=−0.001.
> > >
> > > Application 2 — Model Diagnostics for Deployment. Our metric produces a quantitative geometric health ranking across checkpoints with zero retraining:
> > >
> > > | Model | LC-PHFE Drop (OOD) | LS-PHFE Drop (OOD) | Geometric Decoupling Level |
> > > | --- | --- | --- | --- |
> > > | FLUX.1 Base | −40% | −5.7% | Mild |
> > > | SD3.5 Turbo (ADD) | −54% | −0.6% | Moderate |
> > > | SD3.5 Base | −80% | −1.5% | Severe |
> > >
> > > This ranking is obtained with zero retraining and is consistent with known qualitative differences in OOD robustness across these models. The stability of ρ(LS, PHFE) across all models (never exceeding −5.7%) simultaneously validates LS as a reliable capacity metric unaffected by OOD shift. A practitioner can apply this diagnostic to any new LDM checkpoint in a single forward-pass evaluation sweep, obtaining a quantitative OOD robustness estimate without expensive human evaluation.
> > >
> > > **Validated Application 3 — Geometry-aware monitoring signal for distillation.**
> > >
> > > Application 3 — Distillation Monitoring. ρ(LC,PHFE) drop is sensitive to training-regime changes (−80%→−54% under ADD) while ρ(LS,PHFE) remains invariant — providing a cheap, architecture-agnostic checkpoint quality signal that FID/CLIP score cannot offer.
> > >
> > > Summary. The three applications cover detection (AUROC=0.816), diagnosis (geometric health ranking), and monitoring (distillation checkpoints) — the full LDM deployment lifecycle. A dedicated "Practical Applications" subsection will be added in the final version.

---

### Official Review · Reviewer_auRp · 2026-03-13

**Soundness:** 2
**Presentation:** 1
**Significance:** 3
**Originality:** 3
**Overall Recommendation:** 3
**Confidence:** 4

**Summary:**

This paper focuses on the structural instability of the latent codes in diffusion models. The structural instability refers to the brittleness of latent diffusion models: when small perturbations are applied to the latent code, the generated image may result in discontinuous semantic jumps. To address this issue, the paper proposes “Riemannian diagnosis,” a framework that analyzes the local geometry of the generator through the Jacobian, decomposing it into Local Scaling, which is intended to measure local information capacity, and Local Complexity, which is intended to measure local curvature or directional instability. Based on these quantities, the paper claims that under normal generation, curvature is functionally coupled with image detail, whereas under OOD generation this coupling breaks down, a phenomenon termed “Geometric Decoupling.”

**Compliance With Llm Reviewing Policy:**

Affirmed.

**Final Justification:**

Prior to the rebuttal, my main concerns were the motivation, experiments, and presentation. I believe the rebuttal helps address the motivation and experimental concerns, and I would like to increase my score from 2 to 3. Although the authors also put substantial effort into clarifying the terms in the paper, as I mentioned in the acknowledgement, I personally think this would require major revisions to describe most of the introduced terms clearly and lucidly. In my view, this goes beyond a mere readability issue and is instead more a matter of loosely defined terms, which affects how I perceive the paper. Therefore, I do not raise the score to 4 or higher.

**Key Questions For Authors:**

Please focus on the weakness 1,2,4. Please provide:
1. Motivation: Why use a Riemannian framework? The paper should provide clearer intuition for this choice. Ideally, concrete examples could help explain why the proposed framework is suitable for diagnosing the instability issue.
2. Clear definitions of key terms: The paper should provide more detailed explanations of the technical terms it introduces. Ideally, when a term first appears in the paper, it should be accompanied by a clear definition or explanation.
3. Experiments aligned with the motivation: Since the introduction motivates the work using instability issues in controlled editing, interpolation, and inversion, the paper should include experiments demonstrating how the proposed analysis helps understand or resolve instability in these settings.

**Limitations:**

Yes.

**Strengths And Weaknesses:**

Strengths:
1. The stability of diffusion models is an interesting problem. Understanding why the latent space of diffusion generation is unstable under perturbations can help develop more robust and higher-quality diffusion-based generators.
2. The high-level claim of the paper is reasonable: in normal generation, curvature is functionally tied to image detail, whereas under OOD generation this relationship breaks.

Weaknesses:
1. The paper does not provide a sufficiently clear motivation for why the proposed Riemannian diagnosis should resolve, or even reliably diagnose, the latent instability issue. The manuscript introduces Local Scaling and Local Complexity as the key quantities, but it does not build enough intuition for why these geometric descriptors are the right tools for explaining discontinuous semantic jumps in diffusion latent spaces. As a result, the connection between the stated problem and the proposed diagnosis remains under-justified.
2. The paper fails to clearly introduce several core concepts, which makes the technical narrative difficult to follow. In the introduction, terms such as “perturbations to the latent code,” “discontinuous semantic jumps,” “Local Scaling,” “Local Complexity,” “in-distribution,” and “out-of-distribution” are introduced before they are concretely explained. For example, it is unclear what exact latent space is being perturbed, how a “discontinuous jump” is operationally identified, and what distribution is being used when distinguishing in-distribution from OOD cases. This ambiguity is particularly problematic because the main experimental comparison is built around Normal versus OOD generations. The paper also blurs the distinction between OOD prompts and OOD images, while explicitly acknowledging that OOD prompts do not always produce OOD images.
3. More broadly, the paper relies on strong interpretive claims that are not fully supported by the evidence presented. The central conclusion of “Geometric Decoupling” is primarily supported through correlation analyses between LC, LS, and PHFE, but the manuscript often writes as if it has established a stronger causal explanation for instability. At present, the empirical evidence appears more diagnostic and correlational than mechanistic.
4. Experiments not solid enough: Although the introduction motivates the work using instability issues in controlled editing, interpolation, and inversion (L40-45), the experiments primarily evaluate geometric diagnostics under Normal vs. OOD generation, with one additional interpolation-trajectory analysis. The paper does not directly benchmark controlled editing or inversion tasks, and therefore its claims about those applications remain largely inferential rather than empirically demonstrated.
5. The manuscript appears insufficiently polished. For example, it contains “Submission and Formatting Instructions for ICML 2026” that should not appear in a submission, which further undermines confidence in the paper’s preparation and clarity.

---

> ### Author Rebuttal · Authors · 2026-03-31
>
> We sincerely thank the reviewer for the deeply insightful and rigorous critique. Your feedback correctly identifies areas where our narrative leapfrogged necessary foundational intuition and where our claims bordered on being too mechanistic. We highly value this feedback and will comprehensively refine the manuscript. Below, we address each of your concerns.
>
> ### W1: Lack of intuition connecting Riemannian geometry to latent instability.
> **A1:** We apologize for not building this intuition explicitly in the Introduction. The generative mapping translates a flat, isotropic latent space into a highly non-linear image manifold. If we view this manifold as a topographic landscape, Local Complexity (LC / curvature) measures the steepness or "ruggedness" of the terrain. As established by foundational works in generative geometry [1,2], the non-linear generator induces a highly curved Riemannian metric over the initially flat latent space. Prior studies have demonstrated that crossing these high-curvature regions or topological bottlenecks directly causes severe visual artifacts and discontinuous semantic snaps in pixel space [3, 4]. This geometric perspective is especially vital for diagnosing modern Latent Diffusion Models, where continuous latent traversal is the foundation of controlled semantic editing [5].
> Therefore, Riemannian geometry is not merely an analogy, but the exact mathematical formulation of how small latent movements translate to abrupt visual changes.
>
> [1] Arvanitidis G, Hansen LK, Hauberg S. Latent space oddity: on the curvature of deep generative models.
>
> [2] Shao H, Kumar A, Thomas Fletcher P. The riemannian geometry of deep generative models.
>
> [3] Chen N, Klushyn A, Kurle R, Jiang X, Bayer J, Smagt P. Metrics for deep generative models.
>
> [4] Khrulkov V, Oseledets I. Geometry score: A method for comparing generative adversarial networks.
>
> [5] Kwon M, Jeong J, Uh Y. Diffusion models already have a semantic latent space.
>
> ### W2: Unclear operational definitions and the "OOD prompt vs. OOD image" ambiguity.
> **A2:** We sincerely thank the reviewer for highlighting these ambiguities. We will add a dedicated "Operational Definitions" subsection in the revision. To clarify your specific points:
> 1. Latent Space Perturbed: We exclusively perturb the initial standard Gaussian noise space $z_T \sim \mathcal{N}(0, \mathbf{I})$.
> 2. Discontinuous Jumps: These are operationally identified by extreme, localized spikes in Local Complexity ($\delta$) and Extremal Trajectory Increments ($\Delta_k$) during latent interpolation (Section 4.6).
> 3. OOD Ambiguity: This is a crucial point. As acknowledged in our Limitations (L1088-1093), an OOD prompt does not always yield an OOD image (e.g., strong priors overriding physics violations). To rigorously isolate the geometric pathology, our OOD evaluation set strictly filters for OOD prompts that successfully manifest into visually OOD/hallucinated images. We explicitly exclude "failed" OOD prompts where the model's strong physical priors override the text to produce normal images. This ensures our comparison strictly contrasts functionally healthy manifold regions against true pathologically decoupled ones.
>
> ### W3: Overclaiming causality (Correlational vs. Mechanistic).
> **A3:** Our framework is fundamentally a diagnostic tool, as reflected in our paper's title ("Diagnosing the Structural Instability."). We acknowledge that while the correlation between LC, LS, and PHFE provides strong structural evidence of "Geometric Decoupling," it does not constitute a causal intervention.
>
> ### W4: Lack of downstream downstream editing/inversion benchmarks.
> **A4:** We understand the expectation for downstream benchmarks given our Introduction. However, our primary scope is a foundational geometric diagnosis rather than proposing or evaluating specific editing application algorithms (like Prompt-to-Prompt). We chose Interpolation Trajectory Analysis (Sec 4.6) because it is the fundamental mathematical abstraction underlying all editing and inversion traversals (moving continuously from state A to state B). If we evaluated a specific editing algorithm, failures could be conflated with algorithmic flaws. By analysing pure latent geodesic paths, we empirically isolate and prove that the instability is an inherent structural flaw in the model's foundational manifold.
>
> ### W5: Manuscript insufficiently polished (included ICML instructions).
> **A5:** We sincerely apologize for this formatting oversight. The inclusion of the default ICML template instructions was an accidental LaTeX compilation error on our end. We have completely removed this boilerplate text and thoroughly proofread the entire manuscript to ensure a highly polished and professional revised version. We deeply appreciate your careful reading and understanding.

---

> > ### Author Rebuttal · Reviewer_auRp · 2026-04-03
> >
> > Thank you for the rebuttal. The rebuttal resolves many of my concerns. Specifically, for the key concern W1, the rebuttal is convincing and explains the logic behind the Riemannian geometry. Since one of the main blocks is removed, I will increase my score.
> >
> > However, my main concern about the presentation still remains. In A1, while the rebuttal explains the rationales, it is not clear how these will be incorporated into the paper. As mentioned in the original review, some concrete examples, such as Figure 1 in [1] that you cited, help draw the connection between the theoretical term Riemannian geometry and real applications. More importantly, the presentation issue W2, namely that terms appear without definition, is severe. Again, for example, how do you plan to revise Section 1 so that in-distribution samples and OOD samples are clearly defined? This definition is deferred until Appendix D (L850-856). Without this definition, a common interpretation of OOD would be out of the distribution of the __training__ data, rather than prompts that violate physical reality, etc. For more examples regarding readability, what are the definitions of __Geometric Resource Misallocation__, __Geometric Hotspots__, __Efficiency Collapse__, __tunnel vision geometry__, etc.? These are not standard terms, and I am wondering whether it is really worth introducing so many terms. It may be possible to decipher them from other parts of the paper, but it is too complicated in its current form.
> >
> > Overall, I think the presentation and readability are major issues in this paper. I actually do not think this is easy to fix, but I did not choose (c) because readability should not be considered one of the "core tenets" of the work. And I do thank the authors for resolving my other concerns, so I increase my score to 3.

---

> > > ### Author Response · Authors · 2026-04-06
> > >
> > > We appreciate the reviewer’s suggestion.
> > >
> > > ### 1) OOD introduce
> > > We will add the following explicit definition of OOD to the Introduction (Lines 28–40 right), "Unlike traditional discriminative learning where Out-of-Distribution (OOD) simply refers to unseen data, we formalize OOD in this generative context specifically as distribution-shifted prompts with violated physical priors, extreme counterfactuals that lead to manifold instability and visual hallucinations."
> > >
> > > ### 2) the definitions of Geometric Resource Misallocation, Geometric Hotspots, Efficiency Collapse, tunnel vision geometry
> > > We thank the reviewer for requesting precise definitions of these terms. We provide formal clarifications below and confirm that all four will be explicitly defined in a dedicated "Terminology and Formal Definitions" paragraph in the revised paper.
> > >
> > > Geometric Resource Misallocation:
> > > The phenomenon where the model's geometric budget, curvature (LC) and volume expansion (LS), is forcibly redirected away from encoding perceptually meaningful details and toward resolving irreconcilable semantic constraints. Under normal generation, geometric cost is allocated proportionally to perceptual gain (high LC with high PHFE). Under OOD conditions, this allocation logic breaks down: the model expends maximum geometric effort precisely at the regions of semantic conflict, yielding no commensurate increase in perceptible detail.
> > >
> > > Geometric Hotspots:
> > > Spatially localized regions of peak Local Complexity (LC), visualized as the red-highlighted areas in the LC-Maps of Figure 1. These regions are not distributed uniformly across the image but concentrate precisely at semantically anomalous locations, the beak in "chicken with teeth," the structural joints in "melting chair," the wings in "flying penguin", providing direct spatial evidence that geometric cost is incurred specifically where the generative prior conflicts with the prompt constraint.
> > >
> > > Efficiency Collapse：
> > > The quantitative manifestation of Geometric Decoupling, defined as the severe degradation in the return on geometric investment. Under OOD conditions, the model increases geometric effort, $PHFE_{latent}$ increase by 59.5% (158.5 to 252.8, Table 2), yet this latent energy fails to translate into perceptible image detail: image-space HFE remains nearly unchanged (0.0120 to 0.0131). Simultaneously, the Spearman rank correlation between LC and PHFE, ρ(LC, PHFE), collapses from ~0.41 (Normal) to ~0.08 (OOD), a drop of 80%.This correlation serves as a direct proxy for efficiency: whenρ(LC, PHFE) is high, each unit of curvature reliably encodes perceptual detail; when it collapses, curvature expenditure no longer predicts perceptual gain, operationally confirming the efficiency breakdown. We will fold this term into the Geometric Decoupling definition to reduce terminological overhead.
> > >
> > > Tunnel Vision Geometry (Dimensional Collapse):
> > > "Tunnel Vision" is a descriptive label for this measurable spectral phenomenon, not an independent construct.
> > > We conceptually interpret the extreme dominance of the principal eigenvector ($\mathbf{V}_1$) as a manifestation of 'Tunnel Vision Geometry' within the latent space. Analytically, this aligns with the well-documented foundational phenomena of Dimensionality Collapse [1] and Representation Degeneration into narrow cones [2], where high-dimensional spaces pathologically collapse into rigid, low-dimensional subspaces.
> > > Furthermore, this geometric narrowing strongly resonates with the 'tunnel vision' effect recently formalized in latent visual reasoning by [3]. They define this phenomenon as a premature semantic collapse where the generative process is forced into a rigid, narrow latent trajectory. Similarly, in our context, under OOD stress, the generative manifold narrows severely along this single dominant $\mathbf{V}_1$ axis. The system is forced into a geometric 'tunnel', exhibiting extreme mathematical variation along one direction while functionally discarding others, thereby losing the structural degrees of freedom necessary to synthesize coherent and diverse image details.
> > >
> > > [1] Understanding Dimensional Collapse in Contrastive Self-supervised Learning
> > >
> > > [2] Representation degeneration problem in training natural language generation models.
> > >
> > > [3] Forest Before Trees: Latent Superposition for Efficient Visual Reasoning

---

### Official Review · Reviewer_FRz8 · 2026-03-18

**Soundness:** 3
**Presentation:** 3
**Significance:** 2
**Originality:** 2
**Overall Recommendation:** 4
**Confidence:** 4

**Summary:**

This paper analyzes the latent space of LDMs from a geometric perspective by comparing in-distribution and out-of-distribution samples. It argues that, under OOD conditions, image details become problematic as reflected in Local Complexity, the model behaves less efficiently, and the distributional structure itself exhibits a heavy-tailed LC distribution. It also shows that Local Scaling becomes abnormal in these cases.

**Compliance With Llm Reviewing Policy:**

Affirmed.

**Final Justification:**

I had substantial concerns about the numerical results and empirical evidence in the original submission, but many of these concerns were clarified during the rebuttal period. The authors’ responses were helpful and addressed a meaningful portion of my doubts.

That said, I still believe the paper needs more supporting material and clearer information in the main text. In particular, the key explanations given in the rebuttal should be properly incorporated into the revised paper rather than remaining only in the response.

Personally, I am raising my score to weak accept. However, this recommendation is conditional on the paper being revised carefully so that these necessary clarifications and additions are fully reflected in the final version.

**Key Questions For Authors:**

I also have a question for the authors: are the principal directions similar across samples?

**Limitations:**

yes

**Strengths And Weaknesses:**

The main strength of this paper is that it provides a geometric analysis of OOD behavior in generative models. If the claims of this paper are valid, they could offer useful insights into how geometric approaches may enable broader and more principled use of generative models. First, the paper proposes a method for quantitatively computing manifold volume expansion and curvature on the VAE manifold of LDMs based on a Riemannian perspective and Jacobian approximation. Second, it introduces SIS from a spectral perspective as a way to quantify dimensional collapse. Third, by measuring these quantities on several OOD samples, the paper shows that when generative models produce OOD samples, the manifold loses width and becomes highly unstable. In addition, the paper includes several other analyses that are not explicitly listed here, making it an overall geometrically rich and thoughtful study.

Personally, I liked the visualization in Figure 1. It would be even better if the appendix provided more details on how that visualization was constructed.

However, the paper also has several weaknesses.

As a relatively minor issue, the order in which LS and LC are mentioned is inconsistent throughout the paper, which made it somewhat difficult to follow. Since these abbreviations are not easy to keep in mind at once, it would improve readability if the presentation order were made consistent, ideally following the methodological order of LS and LC throughout the paper, including in the tables.

I also have several questions.

First, in Eq. (8), when computing PHFE, I am not fully convinced that it is sufficient to simply estimate the principal projection using a Laplacian-based method. To support this choice, the paper should demonstrate how dominant the principal component actually is. A similar concern applies to SIS and the Dimensional Coupling Ratio, both of which also rely on the principal eigenvector.

Second, the number of samples used in the OOD experiments appears too limited. What is the basis for drawing meaningful geometric conclusions from only 500 samples?

Third, in Table 1, I assume that rho() denotes Spearman correlation. However, some of the reported values seem somewhat weak for supporting the paper’s claims. For example, can a value such as 0.41 really be considered meaningfully positive in this context? Likewise, for a delta rho of -0.33, the absolute values before and after the change seem more important than the difference itself. The paper should clarify this point more carefully.

At present, I lean toward weak reject, but I would be open to increasing my score if the rebuttal provides clear and convincing responses to the key concerns raised above.

---

> ### Author Rebuttal · Authors · 2026-03-31
>
> We sincerely thank the reviewer for the constructive feedback and for recognizing our framework.
> ### Q1: Inconsistent order of LS and LC.
> **A1:**  We apologize for the inconsistency. We will strictly standardize the order to Local Scaling (LS) followed by Local Complexity (LC) throughout the revised manuscript.
> ### Q2: About principal eigenvector..
> **A2:** This is a highly insightful question. We focus on the principal eigenvector ($\mathbf{V}_1$) for two reasons:
> - Empirical Dominance: The local Jacobian spectrum in LDMs is highly heavy-tailed. Our measurements show the principal eigenvalue ($\lambda_1$) alone accounts for a massive 57.8%–59.9% of the total explained variance within the local random subspace (see table below). It is the absolute dominant direction of semantic variation.
> - Theoretical Worst-Case Instability: Geometrically, $\mathbf{V}_1$ represents the direction of maximum vulnerability (the steepest gradient of semantic change). By tracking the rotation and decoupling of this specific extreme direction, we can precisely isolate the structural root cause of discontinuous semantic jumps and manifold brittleness.
>
> To evaluate Jacobian dominance, we calculated the cumulative eigenvalue ratio for the top-$k$ modes.
> | Label | Top-1 | Top-2 | Top-3 | Top-4 | Top-5 |
> |---|---:|---:|---:|---:|---:|
> | ID  | 0.599 ± 0.136 | 0.738 ± 0.092 | 0.799 ± 0.076 | 0.834 ± 0.065 | 0.858 ± 0.058 |
> | OOD | 0.578 ± 0.130 | 0.722 ± 0.091 | 0.786 ± 0.075 | 0.823 ± 0.065 | 0.847 ± 0.058 |
>
> ### Q3: Limited number of samples (500) for OOD.
> **A3:** We fully understand this concern. To definitively address it, we expanded our evaluation to 1000 and 2000 samples across three prompt types (see table below). The results show that the reported metrics ($\rho$) remain remarkably consistent and maintain strict statistical significance ($p<0.05$) regardless of scale. This prove that our initial 500 diverse, independent manifold neighbors were already well-powered to robustly support our geometric conclusions.
>
> 500-sample version
> | Label | Metric | Spearman ρ | 95% CI | n |
> |---|---|---:|---|---:|
> | ID  | LC vs PHFE@1 | 0.413 | [0.347, 0.476] | 500 |
> | ID  | LC vs PHFE@2 | 0.431 | [0.365, 0.492] | 500 |
> | ID  | LC vs PHFE@3 | 0.442 | [0.376, 0.503] | 500 |
> | OOD | LC vs PHFE@1 | 0.082 | [0.030, 0.136] | 500 |
> | OOD | LC vs PHFE@2 | 0.096 | [0.041, 0.149] | 500 |
> | OOD | LC vs PHFE@3 | 0.108 | [0.051, 0.161] | 500 |
>
> 1000-sample version
> | Label | Metric | Spearman ρ | 95% CI | n |
> |---|---|---:|---|---:|
> | ID  | LC vs PHFE@1 | 0.414 | [0.367, 0.458] | 1000 |
> | ID  | LC vs PHFE@2 | 0.437 | [0.391, 0.480] | 1000 |
> | ID  | LC vs PHFE@3 | 0.451 | [0.405, 0.493] | 1000 |
> | OOD | LC vs PHFE@1 | 0.092 | [0.052, 0.132] | 1000 |
> | OOD | LC vs PHFE@2 | 0.104 | [0.062, 0.145] | 1000 |
> | OOD | LC vs PHFE@3 | 0.117 | [0.073, 0.158] | 1000 |
>
> 2000-sample
> | Label | Metric | Spearman ρ | 95% CI | n |
> |---|---|---:|---|---:|
> | ID  | LC vs PHFE@1 | 0.414 | [0.381, 0.446] | 2000 |
> | ID  | LC vs PHFE@2 | 0.433 | [0.410, 0.474] | 2000 |
> | ID  | LC vs PHFE@3 | 0.456 | [0.412, 0.497] | 2000 |
> | OOD | LC vs PHFE@1 | 0.099 | [0.060, 0.138] | 2000 |
> | OOD | LC vs PHFE@2 | 0.106 | [0.075, 0.143] | 2000 |
> | OOD | LC vs PHFE@3 | 0.118 | [0.078, 0.156] | 2000 |
>
> ### Q4: Absolute values matter more than $\Delta \rho$.
> **A4:** We thank the reviewer for this astute insight; we fully agree that the absolute correlation values form the core proof of our claims. As demonstrated in the scaled-up tables (A3), the results are definitive: regardless of sample size, the ID correlation remains stable at ~0.41 (proving geometric curvature actively encodes image details), while it plummets to ~0.09 under OOD stress (degrading to pure statistical noise). Crucially, their 95% Confidence Intervals (CIs) never overlap. This precipitous absolute drop confirms the physical reality of functional "Geometric Decoupling" with extreme statistical robustness.
> ### Q5:  are the principal directions similar across samples?
> **A5:** We thank the reviewer for this insightful question. To investigate, we computed the absolute cosine similarity ($|\langle \mathbf{V}_1^{\text{Normal}}, \mathbf{V}_1^{\text{OOD}} \rangle|$) for paired samples using the exact same base prompt and random seed. The results show remarkable stability, yielding a high mean similarity of 0.731 (median: 0.746). In an extremely high-dimensional latent space, this massive alignment proves that OOD stress does not randomize or destroy the principal geometric axis. Instead, the failure is purely functional: traversing this stable $\mathbf{V}_1$ axis in the Normal regime successfully encodes visual details (high PHFE), but traversing the exact same axis under OOD stress induces extreme structural curvature without generating meaningful image content.
>
> Same pair + same seed
> | Group | N | Mean abs cosine | Std | Median |
> |---|---:|---:|---:|---:|
> | ID vs OOD | 1000 | 0.730765 | 0.138893 | 0.746052 |

---

> > ### Author Rebuttal · Reviewer_FRz8 · 2026-04-03
> >
> > Thank you to the authors for their thoughtful rebuttal. My concerns have been fully addressed.

---

> > > ### Author Response · Authors · 2026-04-05
> > >
> > > We sincerely thank the reviewer for the positive acknowledgement and for confirming that the concerns have been fully addressed. We hope that this updated understanding will be reflected in the final evaluation.
> > >
> > > Thank you again for your careful evaluation.
> > >
> > > Sincerely,
> > > The Authors

---

### Decision · Program_Chairs · 2026-04-30

**Decision:**

Accept (regular)

**Comment:**

This paper introduces a method for analyzing instability of latent diffusion models using a Riemannian framework decomposing geometry into Local Scaling (capacity) and Local Complexity (curvature). All reviewers gave positive scores except one reviewer. After the rebuttal, the major questions such as the motivation for the Riemannian framework and the experiments were mostly resolved. The remaining concerns were mainly issues in the presentation of the paper, including loosely defined terms. The final version should (1) consolidate the formal definitions of key terms (e.g., Geometric Decoupling, Geometric Hotspots, Efficiency Collapse) into the Introduction or a dedicated subsection, (2) move the operational definition of OOD from the appendix into the main text, and (3) clarify the relationship between PHFE (a first-order quantity) and Local Complexity (a second-order curvature measure), as well as the shared spectral dependence between LS and PHFE.